# Large errors in soil carbon measurements attributed to inconsistent sample processing

Rebecca J. Even[1], Megan B. Machmuller[1,2], Jocelyn M. Lavallee[1,3], Tamara J. Zelikova[1,2], M. Francesca Cotrufo[1,2]

[1]Department of Soil and Crop Sciences, Colorado State University, Fort Collins, CO 80523, USA
[2]Soil Carbon Solutions Center, Colorado State University, Fort Collins, CO 80523, USA
[3]Environmental Defense Fund, 257 Park Ave S, New York, NY 10010

*Correspondence to*: Rebecca Even (rebecca.even@colostate.edu)

**Abstract.** To build confidence in the efficacy of soil carbon (C) crediting programs, precise quantification of soil organic carbon (SOC) is critical. Detecting a true change in SOC after a management shift has occurred, specifically in agricultural lands, is difficult as it requires robust soil sampling and soil processing procedures. Informative and meaningful comparisons across spatial and temporal time scales can only be made with reliable soil C measurements and estimates, which begin on the ground and in soil testing facilities. To gauge soil C measurement inter-variability, we conducted a blind external service laboratory comparison across eight laboratories selected based on status and involvement in SOC data curation used to inform C market exchanges, which could include demonstration projects, model validation and project verification activities. Further, to better understand how soil processing procedures and quantification methods commonly used in soil testing laboratories affect soil C concentration measurements, we designed an internal experiment assessing the individual effect of several alternative procedures (i.e., sieving, fine grinding, and drying) and quantification methods on total (TC), inorganic (SIC), and organic (SOC) soil C concentration estimates. We analyzed 12 different agricultural soils using 11 procedures that varied either in the sieving, fine grinding, drying, or quantification step. We found that a mechanical grinder, the most commonly used method for sieving in service laboratories, did not effectively remove coarse materials (i.e., roots and rocks), thus resulted in higher variability and significantly different C concentration measurements from the other sieving procedures (i.e., 8 + 2 mm, 4 mm, and 2 mm with rolling pin). A finer grind generally resulted in a lower coefficient of variance where the finest grind to < 125 µm had the lowest coefficient of variance, followed by the < 250 µm grind, and lastly the < 2000 µm grind. Not drying soils in an oven prior to elemental analysis on average resulted in a 3.5 % lower TC, and 5 % lower SOC relative to samples dried at 105 °C, due to inadequate removal of moisture. Compared to the reference method used in our study where % TC was quantified by dry combustion on an elemental analyzer, % SIC was measured using a pressure transducer, and % SOC was calculated by the difference of % TC and % SIC, predictions of all three soil properties (% TC, % SIC, % SOC) using Fourier Transformed Infrared Spectroscopy (FTIR) were in high agreement ($R^2$= 0.97, 0.99, 0.90, respectively). For % SOC, quantification by loss on ignition had a relatively low coefficient of variance (5.42 ± 3.06 %) but the least agreement ($R^2$ =

0.83) with the reference method. We conclude that sieving to < 2 mm with a mortar & pestle or rolling pin to remove coarse materials, drying soils at 105 °C, and fine grinding soils prior to elemental analysis are required to improve accuracy and precision of soil C measurements. Moreover, we show promising results using FTIR spectroscopy coupled with predictive modeling for estimating % TC, % SIC, and % SOC in regions where spectral libraries exist.

## 1 Introduction

The potential for carbon (C) sequestration in agricultural soils as an effective climate change mitigation strategy has boosted interest in management practices that increase carbon dioxide ($CO_2$) capture through photosynthesis and its transfer into soil organic matter while also enhancing overall soil health (Bossio et al., 2020). Reliably determining whether shifts in management affect soil organic C (SOC) requires robust sampling, processing, and quantification methods due to the heterogeneous nature of soil. Moreover, changes in SOC are small relative to the standing SOC stocks, making accurate and precise quantification of SOC critical, which requires both reliable bulk density measurements and soil C concentration data. This is especially important now given the expansion of incentives to offset $CO_2$ emissions through purchasing soil C credits (von Unger & Emmer, 2018).

The push to incentivize improved soil management is generally seen as positive but the accuracy and precision of the data that underpin C markets is unresolved (Oldfield et al., 2024). Quantifying changes in SOC stocks by direct sampling requires rigorous approaches, from how samples are collected in the field to laboratory processing and analysis. Major sources of error determining changes in SOC from baseline measurements are the sampling design and location for resampling (Rawlins et al., 2009). For this reason, research focus surrounds uncertainty produced from the soil sampling design (i.e., sampling density) and method, or on the Measurement, Monitoring, Reporting, and Verification (MMRV) protocols (Oldfield et al., 2022), but not specifically on the soil processing methods. Yet, for reproducibility and comparability of SOC stocks across spatial and temporal scales, sample preparation is considered one of the most important quantification steps (Theocharopoulos et al., 2004). The variability produced due to soil processing alone has yet to be explored and often goes without scrutiny in C quantification protocols such as those for market applications. Reducing variability and developing clear guidance on soil processing best practices can improve MMRV approaches and credibility of C market programs, policy, corporate sustainability efforts, and overall climate progress. Additionally, these same data underpin the calibration and verification of soil biogeochemical models used to predict how changes in management affect SOC dynamics, making it more imperative that we use the best approach to obtain accurate SOC data across soil testing laboratories worldwide.

Soil testing labs can elect to participate in quality assurance and quality control (QA/QC) certification programs that promote their data as high quality. For example, the North American Proficiency Testing (NAPT) Program is offered in the United States (U.S.), with over 130 NAPT certified labs. Participating labs are sent soil samples either quarterly or biannually and the data generated by each lab is subjected to a blind and double-blind statistical evaluation. Values within +/- 2.5 times the median absolute deviation (MAD) units of the median (S890 North American Proficiency Testing program oversight

committee, 2020) are considered acceptable. However, labs receive soil already processed using the same methods. Hence, the precision of the quantification method (i.e., instrumentation) being used to measure soil C is the sole focus of the testing and certification. There is no focus on soil processing, thus no evaluation of how soil C data can vary across laboratories due to differences in their soil processing methods. And while several standards exist that guide methodologies for soil analysis of SOC (e.g., the International Organization for Standardization (ISO) 23400:2021(en)), divergence of methods occurs as the guidelines are not followed by every lab, especially for soil preparation procedures prior to elemental analysis.

Typically, in research laboratories, soil analyses are "conventionally" performed on soils passed through a 2 mm sieve (Bernoux & Cerri, 2005) with large (> 2 mm) coarse materials like rocks and roots removed since they are not considered part of the fine soil (Brady and Weil, 1996) and affect soil C estimates if left in the sample. Fine plant materials that are larger than 2 mm but still pass through a 2 mm sieve are often hand-picked using tweezers. Processing soils involves breaking soil aggregates at natural planes of weakness (Arshad et al., 1997) using hand manipulation as a gentle approach during sieving (e.g., Clement and Williams, 1958; Diaz-Zorita et al., 2002). However, because this process is time consuming and cropland soils often have low coarse material content, in soil testing facilities processing protocols for the assessment of cropland soil fertility have typically not included careful aggregate breaking and sieving to remove coarse materials. Compared to conventionally managed croplands, croplands managed using a regenerative practice, like the addition of certain perennial crops (i.e., alfalfa), typically have more coarse materials deeper in the soil profile as more root biomass is incorporated at depth (Fan et al., 2016). Additionally, grasslands are a main target of soil C programs, and their soils often have abundant and deep root content as well as gravel (Bai and Cotrufo, 2024). Thus, it's important to consider how coarse materials in these soils may affect C estimation.

Variation in sieving approaches exists, from hand sieving methods where soils are either sieved fresh (i.e., field moist) to 8 mm initially and then 2 mm sieved once air-dried (Mosier et al. 2021), sieved air-dried through a 4 mm sieve (Syswerda et al., 2011), or sieved using a customized rolling pin to break up aggregates to pass through a 2 mm sieve (Soil Survey Staff, 2022). Coarse materials (i.e., roots, aboveground litter, and rocks) should be removed and quantified at each sieving phase. Most soil processing methods include a step to sieve air-dried bulk soil and commercial service soil testing labs often utilize a mechanical flail or disk grinder (Garcia et al., 2022) to increase throughput over hand sieving. Based on a survey of 51 soil testing labs, we found that over 70 % use some type of mechanical grinder to sieve (Supplemental Table S1). This is achieved by pouring air-dried bulk soil into a grinder cavity powered by electricity with stainless steel flailers (or something similar) to break soil aggregates before passing through a 2 mm screen. The material that passes through the 2 mm screen is considered fine soil and utilized for analyses. It is possible that not all soil falls through the 2 mm sieve on the first pass and, because of the destructive nature of the mechanical grinder, that some coarse material is broken up enough to then pass through the sieve. While some labs may only pass the soil through once, others may pass it through several times to ensure all aggregates are broken but there is no standardized procedure that outlines how many times the soils should be processed with mechanical grinders and how coarse materials are considered.

Subsequent soil processing steps and the specific C quantification approach may all potentially affect the final soil C concentration measurement or estimate. After sieving, soil samples can be finely ground to improve homogenization or left unground. If using a mechanical grinder, it's typical that soils are considered ground, but given this is to < 2 mm only, it may be inadequate in terms of homogenization. As C is estimated on a small subsample (e.g., less than a gram) its homogenization at smaller scales than 2 mm may be critical to obtaining a representative value. Additionally, soil C is measured as a concentration, i.e. per unit of soil mass, therefore the presence of water in soil will affect soil mass, diluting C concentration estimates. To further remove moisture from air-dried samples, soils can be oven-dried, but the temperature used varies between 45 °C and 105 °C (Supplementary Table S2). Some labs apply a moisture correction based on residual moisture if the soils are not oven-dried but in other cases labs neither oven-dry nor apply a moisture correction.

After processing, soil samples are measured for C content via one of three common methods: (1) dry combustion in an elemental analyzer (EA; Nelson & Sommers, 1996), where the soil sample is combusted under a flow of $O_2$ at a temperature of 950-1100 °C, converting all C in the soil to $CO_2$ that is then quantified by a Thermoconductivity detector (TCD; Bisutti et al., 2004); (2) loss on ignition (Storer, 1984) where soil is heated to 360 °C or higher to oxidize the organic matter in the soil which is calculated by the sample mass difference before and after heat treatment, and % SOC is obtained using a conversion factor; (3) Fourier-transformed infrared spectroscopy (FTIR) (Reeves, 2010; Goydaragh et al., 2021) where samples are scanned in the mid-infrared region. The produced spectra are then coupled with predictive models that must be well trained to produce accurate soil C estimates.

For typical midwestern U.S. soils with neutral to basic pH, C may also be present in inorganic forms (i.e., carbonates) requiring the further separation of total C (TC) into soil inorganic (SIC) and organic C (SOC). Thus, all soils with pH at or greater than 6.95 (Soil Survey Staff, 2022) should be tested for the presence of SIC or treated as if SIC is present. The test for SIC is commonly performed by applying an acid solution to the soil sample and confirming whether the sample has carbonates based on effervescence (i.e., fizz test), as the acidity converts the carbonates to $CO_2$. After a positive fizz test, soil C requires the quantification of both SOC and SIC, whose sum makes TC. This is commonly done in one of two ways: (1) SOC is quantified by fumigating soils with acid (Harris, 2001) to remove the SIC and running samples on the EA for the estimation of % SOC, after which SIC is estimated by subtracting SOC from TC measured on a corresponding unfumigated sample; (2) SIC is quantified using a pressure transducer (Sherrod et al., 2002) that measures the pressure of the $CO_2$ released by the reaction of applied HCl and calcium carbonate ($CaCO_3$) from the sample, then SOC is calculated by subtracting SIC from TC measured on an EA. Since one or the other (SOC or SIC) entity is calculated rather than directly measured, both estimates are inaccurate if the measured data are not accurate. Additionally, accurately measuring either SOC or SIC becomes more difficult on the high end of the SIC concentration spectrum (McClelland et al., 2022; Stanley et al., 2023). Differences in SOC and SIC quantification methodologies have been explored (Bowman et al., 2002; Sleutel et al., 2007; McCarty et al., 2010; Farmer et al., 2014; Wang et al., 2012; Apesteguia et al., 2018; Leogrande et al., 2021) and can have large impacts on calculated SOC. However, to our knowledge, there are no studies that directly compare the different sieving, grinding, and drying steps of soil

processing, and final measurement methods for their potential impact on TC, SOC and SIC quantification in an extensive and factorial design.

To understand how alternative soil processing procedures and quantification methods affect soil C estimates, we designed an experiment to assess the individual effect of common alternative approaches for sieving, grinding, drying and quantification methods described above for TC, SIC, and SOC concentrations, applied to 12 diverse agricultural soils. In addition, we conducted a blind external service laboratory comparison, sending subsamples to eight popular commercial labs that provide soil carbon estimation.

We expected that sieving, fine grinding, and drying would all affect soil C data. We hypothesized that the 8 + 2 mm sieve method would have the lowest variability because it would remove the most coarse material. We expected higher SOC concentrations where coarse materials left in the soil were organic (e.g., roots) as they would add to the total C, and lower SOC concentrations where coarse materials left in the soil were rocks, as they would add to the soil mass. We expected soils that were not finely ground would have higher variability for lack of homogenization, and that protocols which did not oven-dry the sample would produce more variable and lower % C values. Lastly, we expected that loss on ignition would perform worse than the other quantification methods for SOC and that FTIR would be a viable alternative, specifically for predicting % SIC. For the external laboratory comparison, we expected that there would be some variability among laboratories, but that data would be within a reasonable range (e.g., values within +/- 2.5 times the MAD units of the median by NAPT standards) since laboratories were selected based on status and involvement in SOC quantification for carbon markets.

## 2 Materials & Methods

### 2.1 Soils and service laboratory comparison

Soils were selected from 12 agricultural sites in central U.S. to represent a range of textures, SOC concentrations, presence and proportion of SIC, and coarse material (i.e., plant and rock) contents (Table 1). The pH was determined using a 1:1 ratio of soil to deionized water. Texture was determined after shaking 40 g of soil in 5 % sodium hexametaphosphate solution for 18 hours, wet sieving sand > 53 µm, and using a hydrometer to determine silt and clay content. Texture classes were defined according to the soil texture calculator created by the Natural Resources Conservation Service U.S. Department of Agriculture.

**Table 1:** Properties, land use, and state and county of sampling for the 12 agricultural soils used for this study. Soils are identified by capital letters. Concentrations of total carbon (% TC), soil organic carbon (% SOC) and soil inorganic carbon (% SIC) are reported as averages ± standard deviations (n=5). Proportions of rock and plant materials were categorized as low (below the 25 % quartile), medium (in the 25 % - 75 % interquartile), and high (above the 75 % quartile). All data reported were obtained using the refence protocol (Fig. 1). Soils sent to the external laboratories for soil carbon analyses are indicated with an asterisk (*). Not applicable (NA) for % SIC indicates no presence of SIC in the soil.

| Soil | pH 1:1 | Texture | % TC | % SOC | % SIC | State | County | Land use | rocks | plant material |
|------|--------|---------|------|-------|-------|-------|--------|----------|-------|----------------|
| A | 6.88 | clay loam | 4.26 ± 0.3 | 3.58 ± 0.28 | 0.68 ± 0.04 | Iowa | Fayette | Annual cropland | medium | medium |
| B* | 7.43 | silty clay loam | 3.11 ± 0.36 | 2.55 ± 0.45 | 0.56 ± 0.11 | Colorado | Delta | Grazed pasture | medium | high |
| C* | 6.56 | sandy clay loam | 1.31 ± 0.09 | 1.31 ± 0.09 | NA | Wyoming | Albany | Rangeland | high | medium |
| D* | 7.58 | clay loam | 1.7 ± 0.06 | 0.85 ± 0.05 | 0.86 ± 0.02 | Colorado | Otero | Annual cropland | low | low |
| E | 5.25 | silt loam | 1.1 ± 0.05 | 1.10 ± 0.05 | NA | Colorado | Kit Carson | Annual cropland | medium | medium |
| F | 5.2 | loam | 1.07 ± 0.05 | 1.07 ± 0.05 | NA | Colorado | Phillips | Annual cropland | high | medium |
| G | 6.75 | sandy loam | 1.38 ± 0.08 | 1.38 ± 0.08 | NA | Colorado | Kit Carson | Annual cropland | medium | high |
| H* | 7.56 | silt loam | 7.21 ± 0.04 | 2.34 ± 0.45 | 4.86 ± 0.42 | Kansas | Riley | Perennial grazing | high | high |
| I | 6.78 | loam | 1.41 ± 0.1 | 1.41 ± 0.10 | NA | Illinois | Mercer | Annual cropland | low | medium |
| J* | 6.1 | silty clay loam | 3.3 ± 0.16 | 3.30 ± 0.16 | NA | Iowa | Scott | Perennial cropland | low | low |
| K | 7.24 | loam | 2.5 ± 0.19 | 2.33 ± 0.19 | 0.17 ± 0.01 | Iowa | Jasper | Annual cropland | medium | low |

| L | 7.53 | loam | 2.98 ± 0.29 | 2.55 ± 0.26 | 0.43 ± 0.09 | Iowa | Jasper | Annual cropland | medium | medium |

Land cover at these sites includes cropland, pasture, rangeland, and tall grass prairie. To collect a relatively uniform sample and avoid a strong influence of spatial heterogeneity, soils were collected by spade from a small area, roughly 50 cm x 50 cm. The intention of this sampling procedure was not to obtain a sample that represented the field site or a large area (e.g. on the hectare scale), rather to collect enough soil with unique (relative to other sites) and uniform properties (within the collected soil) to use for the laboratory procedure comparison. Once in the lab, soils were stored in a 4 °C refrigerator until further use. Each field moist soil was homogenized to the best of our ability. We sought to minimize variability by spreading the entire sample out on kraft paper, flipping the soil over itself twice, and collecting soil from various parts of the kraft paper to ensure representative subsamples. The kraft paper used had no water-soluble C. Further, we subsampled for all replicates, including those sent to external laboratories and those retained at Colorado State University (CSU) for the protocol comparison in the same way to minimize differences across laboratories and methods due to inherent soil heterogeneity.

A subset of soils was selected for the external service laboratory comparison (soils B, C, D, H, and J; Table 1), with the aim of capturing a broad range of coarse material contents, soil textures, and SOC and SIC concentrations while balancing costs. For soils B, C, D and J, a subsample of each homogenized field moist soil was further separated into two subsamples: (1) a subsample that was air-dried and (2) a subsample that was maintained field moist at 4 °C. These two subsamples (from here on referred to as air-dried or field moist) from each soil were sent to the eight external service laboratories. For soil H, we sent an air-dried sample only as we did not have enough soil to include a field moist sample. The processing and quantification methods used by each external laboratory are described, to the best of our knowledge, after reviewing the standard operating procedures and communicating directly with lab personnel, in Supplemental Table S2.

### 2.2 Experimental design for testing soil processing and carbon quantification protocols

To test the effect of sieving, grinding, drying and final quantification methods on estimates of TC, SIC, and SOC concentrations, we processed and analyzed the 12 soils described in Table 1 following eleven different protocols (Fig. 1). Each protocol was replicated five times per soil for all 12 soils. We considered the methods used in the Soil Innovation Laboratory at CSU as the reference (R) where all protocols deviated from R for one step to enable the evaluation of the effect of each individual step on the estimation of TC, SIC and SOC concentrations.

### 2.3 Soil processing

To test the effect of sieving, each field moist soil was homogenized and split into five representative replicates as described above. The replicates from each soil were further split into one batch that was air-dried, and one batch maintained

moist at 4 °C. The latter was processed according to the R sieving protocol (Fig.1), by passing field moist soil through an 8 mm sieve to remove coarse material. A 25-30 g subsample was dried at 105 °C to calculate soil moisture and the remaining soil was air-dried. Once air-dried, all soil was passed through a 2 mm sieve to remove additional coarse materials not considered soil (> 2 mm): any aggregates remaining on the sieve were gently broken with a mortar and pestle until they could pass through the sieve. All coarse materials were dried at 60 °C and weighed. Soils from the air-dried batch were split into three representative subsamples to test alternative sieving methods S1, S2, and S3 (summarized in Fig. 1). The S1 soils were passed through a 4 mm sieve, following a protocol used by the Kellogg Biological Station at Michigan State University (i.e., in Syswerda et al., 2011), removing any coarse materials larger than 4 mm, gently breaking aggregate structures using the same method as R (mortar and pestle) until all soil passed through. The S2 soils were processed using a rolling pin, following a modified protocol utilized by the Kellogg Soil Survey Laboratory (Soil Survey Staff, 2022). Soils were first poured over the 2 mm sieve and coarse fragments remaining on the sieve were removed.

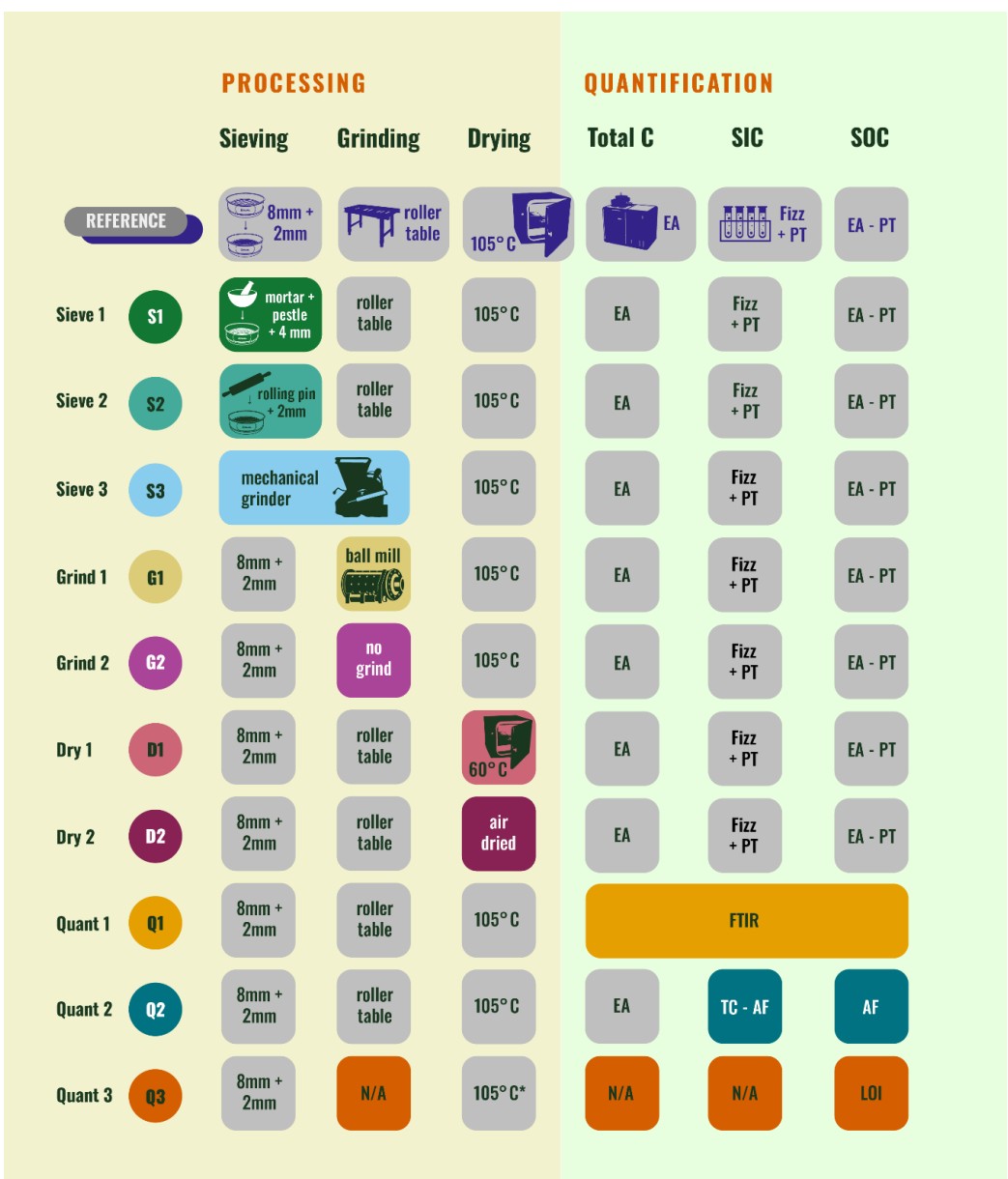

**Figure 1:** The procedural variations for sieving, grinding, drying, and quantification methods of total carbon (Total C), soil inorganic carbon (SIC), and soil organic carbon (SOC) concentrations. Sieving variations included the Reference (R; 8 + 2 mm), S1 (4 mm), S2 (2 mm with rolling pin), and S3 (mechanical grinder). Grinding variations include R (roller table grind to < 250 µm), G1 (ball mill to < 125 µm), and G2 (no grind; < 2000 µm). Drying variations include R (105 °C), D1 (60 °C), and D2 (air-dried only). For the quantification of % TC, dry combustion by elemental analyzer (R; EA) and Fourier transformed infrared spectroscopy (Q1; FTIR) were tested. Quantification for % SIC was tested using a pressure transducer (R; PT), FTIR (Q1), and acid fumigation (Q2; AF) where SIC is calculated by subtracting SOC from TC (EA with no AF) . SOC quantification

procedures included subtracting SIC (PT) from TC (EA) concentrations (R), FTIR (Q1), AF (Q2), and loss on ignition (Q3; LOI) where mass loss was based on soils dried at 105 ºC prior to the muffle furnace (*).

Aggregates left on the sieve were gently crushed on kraft paper with the rolling pin until all soil passed through the sieve. The S3 soils were processed using a DC-5 Dynacrush mechanical flail grinder which breaks soil aggregates in a chamber using stainless-steel flailers and allows soil to fall through a 2 mm screen as aggregates are broken. The whole sample was poured into the cavity for the first pass. Large coarse fragments left on the 2 mm screen were removed and contents (i.e., aggregates) remaining on the screen were passed through again. This process was repeated three times. Soil aggregates that did not get crushed to pass through the 2 mm screen were collected and quantified separately from coarse materials. All coarse material for S1, S2, and S3 methods were air-dried and weighed. Soil samples processed according to the R, S1, S2 sieving procedures were finely ground and dried at 105 °C. S3 soils were considered "ground" by the Dynacrush as is standard procedure and were dried at 105 °C. Thus, the S3 is the only tested protocol which differed from R for two steps, sieving and grinding, as the mechanical grinding method substitutes them both.

Soils were defined as having high, medium, or low rock or plant contents (Table 1) according to the amount of coarse material removed using the 8 + 2 mm (R) sieving procedure and the boxplot function in RStudio. The categories were based on the interquartile ranges where plant or rock proportion by mass less than the 25[th] percentile was classified as low, medium fell in the interquartile range (between the 25[th] and 75[th] percentile), and high was proportion by mass greater than the 75[th] percentile.

To test the effect of fine grinding, soils which had all been sieved and dried according to the R procedure were either pulverized using a roller table with three metal bars (R; Arnold and Schepers, 2004) for 18 hours until soils were < 250 μm, a ball mill (Planetary Mill Pulverisette 5; G1) for 2 minutes, pulverizing soils to < 125 μm or not ground (G2) beyond the 2 mm sieve (< 2000 μm). To test the effect of drying soils at different temperatures, soils sieved and finely ground according to the R procedure were dried in the oven at 105 °C (R), at 60 °C (D1), or left to air-dry (D2).

All soils from the R to D2 processing protocols were analyzed for % TC, % SIC and % SOC according to the R method described below, so that their comparisons could specifically test the effects of the methodological step for which they differed on soil C values.

**2.4 Soil analyses**

To assess soil C quantification methods, soils that had been sieved according to the R procedure were analyzed using methods Q1, Q2 and Q3 (summarized in Fig. 1). R soils were analyzed for % TC by dry combustion (Nelson & Sommers, 1996) on an elemental analyzer (EA) for % SIC by pressure transducer (modified from Sherrod et al., 2002), and % SOC was calculated by subtracting % SIC from % TC. The dry combustion method (R; EA) is considered the most accurate method for total C quantification (Yeomans & Bremner, 1988; Bisutti et al., 2004) so it is often used as a reference (Leong & Tanner,

1999; Bisutti et al., 2004) against other quantification methods. SIC concentration was determined using the pressure transducer as the R method because, in our experience, it is a more efficient and cost-effective way to quantify SIC compared to acid fumigation (TC – SOC) where soil samples must be analyzed twice on the EA. For Q2, % TC was first measured on the EA, % SOC was measured on the EA after acid fumigation to remove carbonates (Harris et al., 2001) and SIC was calculated by subtracting % SOC from % TC. The Q1 method used FTIR to predict TC, SIC, and SOC concentration, following the analytical procedure described below. For the Q1 method, we tested four different grinding approaches that produced different sample particle sizes (no grind to < 2000 µm, roller table grind to < 250 µm for 18 hours, roller table grind to < 180 µm for 48 hours, and ball mill pulverization to < 125 µm) as the effect of soil particle size may play a lesser role on FTIR accuracy than previously reported (Sanderman et al., 2023). The results of variation in particle size are reported in Supplemental Figure S1. The test suggested that the < 180 µm grind was sufficient for FTIR scanning, which was the particle size of the samples used to build the NRCS-KSSL spectral library (Seybold et al., 2019) and models used in this study. Thus, we compared the Q1 < 180 µm protocol to the other quantification methods. Lastly, for the Q3 method, % SOC was estimated using loss on ignition (LOI), following Storer (1984). Briefly, soil (10 g) was heated to 360 °C for 2 hours in a muffle furnace to oxidize organic matter. The difference in mass was calculated on soils dried at 105 ºC and then equated to the organic matter content and a conversion factor of 0.58 (Pribyl, 2010) was used to estimate % SOC.

## 2.5 Analytical procedure

### 2.5.1 Quantification of carbon by elemental analyzer

We used a vario ISOTOPE cube elemental analyzer (Elementar, Germany) housed at the Soil Innovation Laboratory at Colorado State University to measure % TC and % SOC after acid fumigation for the Q2 procedure only. The EA was drift corrected every day before measurement, using an acetanilide standard (Sigma Aldrich) with 71.09 % C in three replicates. Following the three acetanilide standards, three replicates of an internal soil standard, with % TC comparable to the samples, were run to ensure data reliability. The mass of soil used was related to its % TC where approximately 30 mg of sample was used for low % TC soils and 10 mg was used for soils considered to have medium % TC. The internal standard and a duplicate sample were run every 10 samples to continuously check for drift. The manufacturer reports a precision of < 0.1 % TC, and we observed an average standard deviation of 0.04 % TC across all soil standards run in this data set.

### 2.5.2 Fourier transformed infrared spectroscopy

Samples were scanned in the mid-infrared region as described by Leuthold et al. (2024). For the analyses, roughly 40 mm$^3$ of sample was pressed down using a metal rod to fill a 6 mm well in a 96 well plate. This ensured an even surface as soil was scanned 32 times at a resolution of 4 cm$^{-1}$ from 7500 to 600 cm$^{-1}$ on a VERTEX 70/HTS-XT INVENIO-R FT-IR (Bruker Optics Inc., Billerica, MA, USA). A gold background was scanned before every sample to correct for potential fluctuations and interference of $CO_2$ and $H_2O$. Samples were scanned in quadruplicate and then analyzed for similarity. The first scan of each sample was considered the reference, and the scans were accepted when similarity passed with a correlation of 0.996 or

higher for the other three scans, otherwise all four scans were repeated. For predicting % TC, % SIC, and % SOC, calibration models were built using the USDA NRCS National Soil Survey Center-Kellogg Soil Survey (NSSC-KSSL) spectral library coupled with partial least squares regression in the OPUS software (OPUS version 8.5, Bruker Optik GmbH 2020) as described in detail by Seybold et al. (2019). The calibration models were developed separately by soil property and geographical region. Spectra were trimmed to the mid-infrared region from 4000 to 600 cm$^{-1}$ and calibration spectra were mean centered with redundancies removed using principal component analysis and outliers removed based on ANOVA of residuals in OPUS. Details for the geographical boundaries, spectral pre-processing, R², and root mean square error of prediction for each model can be found in Supplemental Table S3.

### 2.6 Data analysis

All statistical analyses were performed in RStudio version 4.3.1 (R Core Team, 2024). Linear mixed models were used (*lme4* package; Bates et al., 2015), with soil and procedure as interacting fixed effects and soil replicate as a random effect, to compare % TC, % SOC, % SIC, and % coarse material (plant and rocks) across procedures. A log-transformation was utilized to fulfill the assumptions of normality and equal variance except for % plant material and rocks where a square root transformation was used to accommodate values of 0. The coefficient of variance (CV) was calculated by sd/mean*100 for each soil*procedure combination and then averaged over soil for each procedure. Simple linear models were used to test the effect of procedure on CV. Pairwise comparisons were made using the *emmeans* (Lenth, 2024) package with a Tukey adjustment. Differences with a p-value < 0.05 were considered significant.

### 3 Results

### 3.1 External lab comparisons of soil carbon analyses

Within a given lab, reported values for the same soil (sent either as air-dried or field moist) varied by up to 4.62 % TC, 4.06 % SIC, and 1.45 % SOC. (Fig. 2; Supplemental Table S4). Lab III and Lab IV both reported absolute differences of nearly 2 % less TC in the field moist sample than in the air-dried sample of soil B. Lab III reported a much higher % SOC value in the air-dried sample (4.07 % SOC) than the field moist sample (2.62 % SOC) from soil B, a soil with 12 % soil moisture and high root material (Table 1). However, in some cases, labs reported the same or similar values either air-dried or field moist. For example, Lab VII, reports no difference in % SOC while Lab I only detected a 0.01 % difference between the air-dried and field moist samples sent from soil B. Lab VI reported differences of < 0.1 % TC for all soils while Lab I reported differences in SOC of < 0.1 % for all soils between the air-dried and fresh subsamples sent (Fig. 2).

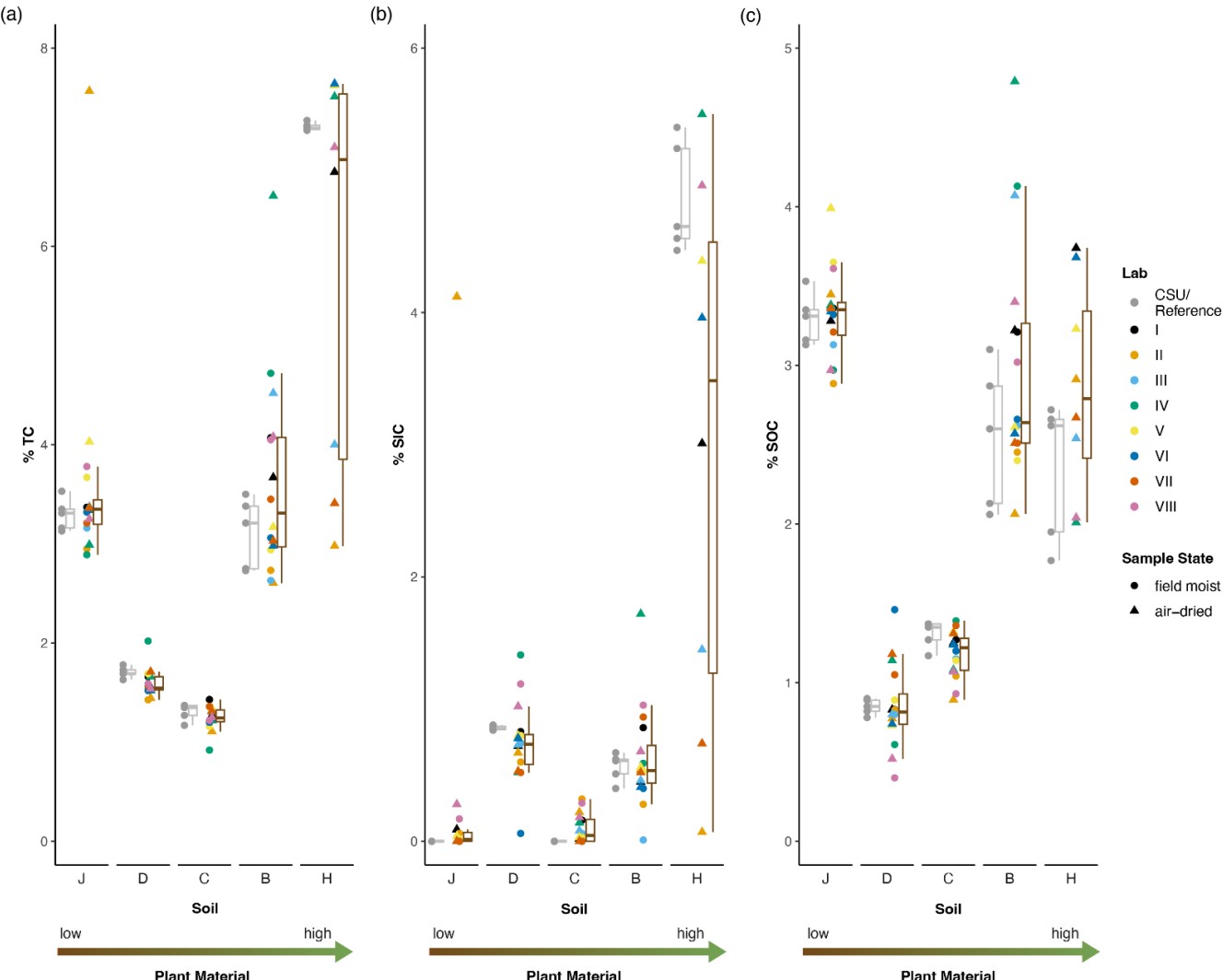

**Figure 2:** The distribution of total carbon (TC; panel a) soil inorganic carbon (SIC; panel b) and soil organic carbon (SOC; panel c) concentrations from eight service soil testing laboratories and our reference method (CSU/Reference). Box plots report the median, first and third quartiles for values from all soils analyzed at service soil testing laboratories (brown boxplot; field moist and air-dried combined for soils B, C, D, and J; n=16) and by the CSU/reference method (grey boxplot; n=5). Whiskers extend to the upper and lower data point that are within 1.5 times the interquartile range. One sample from soil H was sent to each lab (n=8). For descriptions of the soil properties, CSU/Reference method and methods used by the external service soil testing laboratories refer to Table 1, Figure 1 and Supplemental Table S2, respectively.

Differences also occurred in estimates across labs (Fig. 2). Even in soil with lower % TC (i.e., soils C and D) where less variability occurred generally, we still observed notable differences in % SOC where Lab VI reported 1.46 % SOC and Lab VIII reported 0.40 % SOC for soil D. We observed an extreme outlier of 4.12 % SIC from the air-dried sample of soil J reported by Lab II, a 118-fold difference than the average across all other labs of 0.035 ($\pm$ 0.016) % SIC. For soil H, we could not make comparisons of air-dried vs. field moist soil C data since we only sent an air-dried sample. However, the range across laboratories for % SIC at soil H is remarkable. We observed an average of 3.72 ($\pm$ 1.83) % SIC across all labs for soil H where Lab II reports 0.07 % SIC while Lab IV reports 5.5 % SIC (Fig. 2). The distributions of % SIC data were tighter for soils with lower % TC but we still observed considerable distributions for soil D looking at % SIC where Lab VI reports 0.06 % SIC while Lab IV reports 1.41 % SIC (Fig. 2). Overall, variability in C measurements between labs and within the CSU/reference method was highest for soils B and H.

## 3.2 Assessment of the effect of soil processing procedures on soil carbon measurements

### 3.2.1 Effects of sieving on soil carbon measurements

The amount of coarse material removed from the soil samples was dependent on the procedure used for sieving, as expected. Overall, sieving by mechanical grinder (S3) removed less plant material than the other procedures, especially in soils characterized by a higher proportion of plant material by mass (Table 1; Fig. 3). There were no differences in % plant material removed for the soils defined as having low plant material between any of the sieving procedures. In soils with medium plant material, the mechanical grinder (S3) removed less plant material than the 8+2 mm sieve (R) and 4 mm sieve (S1) in four and three out of six soils, respectively. The 8 + 2 mm (R) and 4 mm (S1) sieving procedures removed significantly more plant material than the mechanical grinder (S3) in all soils defined as having high plant material while the 2 mm sieve with rolling pin (S2) removed more in two thirds of the high plant soils (Supplemental Table S5). Averaged across soils, the 8 + 2 mm (R), 4 mm (S1), and 2 mm rolling pin (S2) sieving procedures removed 69 %, 66 %, and 59 % more plant material than the mechanical grinder (S3). The 2 mm sieve with rolling pin (S2) removed less plant material than the 8 + 2 mm sieve (R) in three out of six soils with medium plant material and one third of soils with high plant material.

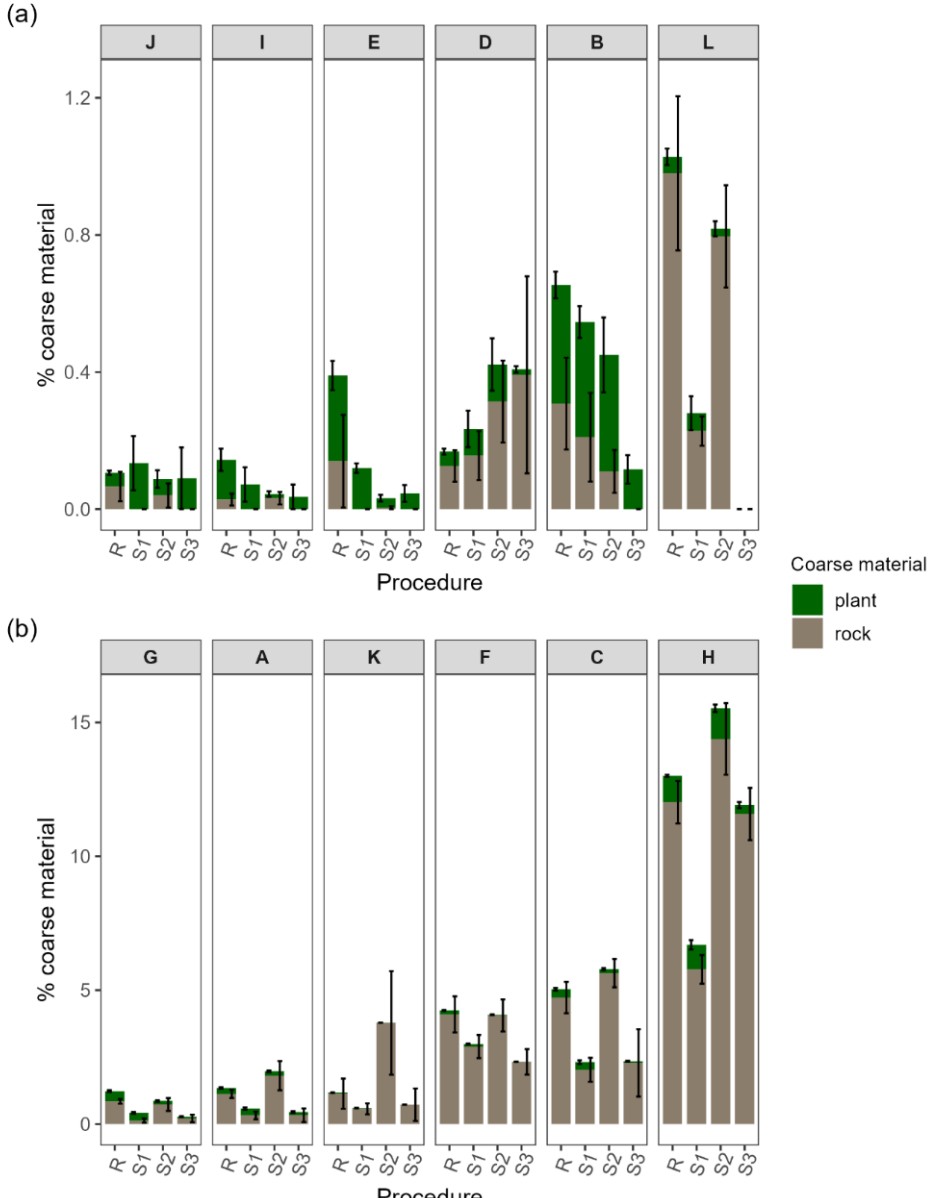

**Figure 3:** A stacked bar graph illustrating the proportion of coarse material removed from the total soil mass with four different sieving procedures: R (8 + 2 mm), S1 (4 mm), S2 (2 mm with rolling pin), and S3 (mechanical grinder) described in Figure 1. Stacked bars represent the mean (± standard error; n=5) of coarse material identified as plant (top; green) or rock (base; beige). Letters refer to soils as described in Table 1. Panel a (top) includes soils with low coarse material (up to 1% on average), and Panel b (bottom) includes soils with more than 1% coarse material

There was a significant interaction of soil and procedure for the effect of sieving on rock removal (Supplemental Table S5). We observed no differences in rock removal based on sieving procedure in any of the soils with low rock content. However, in soils with medium rocks the mechanical grinder (S3) removed less than the 8 + 2 mm (R) and 2 mm with rolling pin (S2) sieve in three out of six soils and one third of soils considered to have high rock content. In the soil with the highest rock content (H), all sieving procedures removed more rock than the 4 mm (S1) sieving procedure. Moreover, sieving only to 4 mm (S1) removed less rock than sieving to 8 + 2 mm (R) in two out of six soils with medium rock content.

An interaction of sieving procedure and soil (Supplemental Table S6) revealed that in a quarter of the soils, the mechanical grinder (S3) resulted in a lower % TC than the 8 + 2 mm (R) and 4 mm (S1) sieving procedures. Compared to the 2 mm sieve with rolling pin (S2), the mechanical grinder (S3) had a lower % TC in two out of twelve soils. There were no significant differences in % TC between the 8 + 2 mm, 4 mm, and 2 mm with rolling pin sieving procedures for any of the soils (Supplemental Fig. S2 & Table S6). For % TC, the 2 mm sieve with rolling pin (S2) had a lower CV on average compared to the mechanical grinder (p=0.033; Supplemental Fig. S3). Sieving procedure had a significant impact on % SIC (p=0.011) but there was no interaction between sieving procedure and soil (Supplemental Table S6).

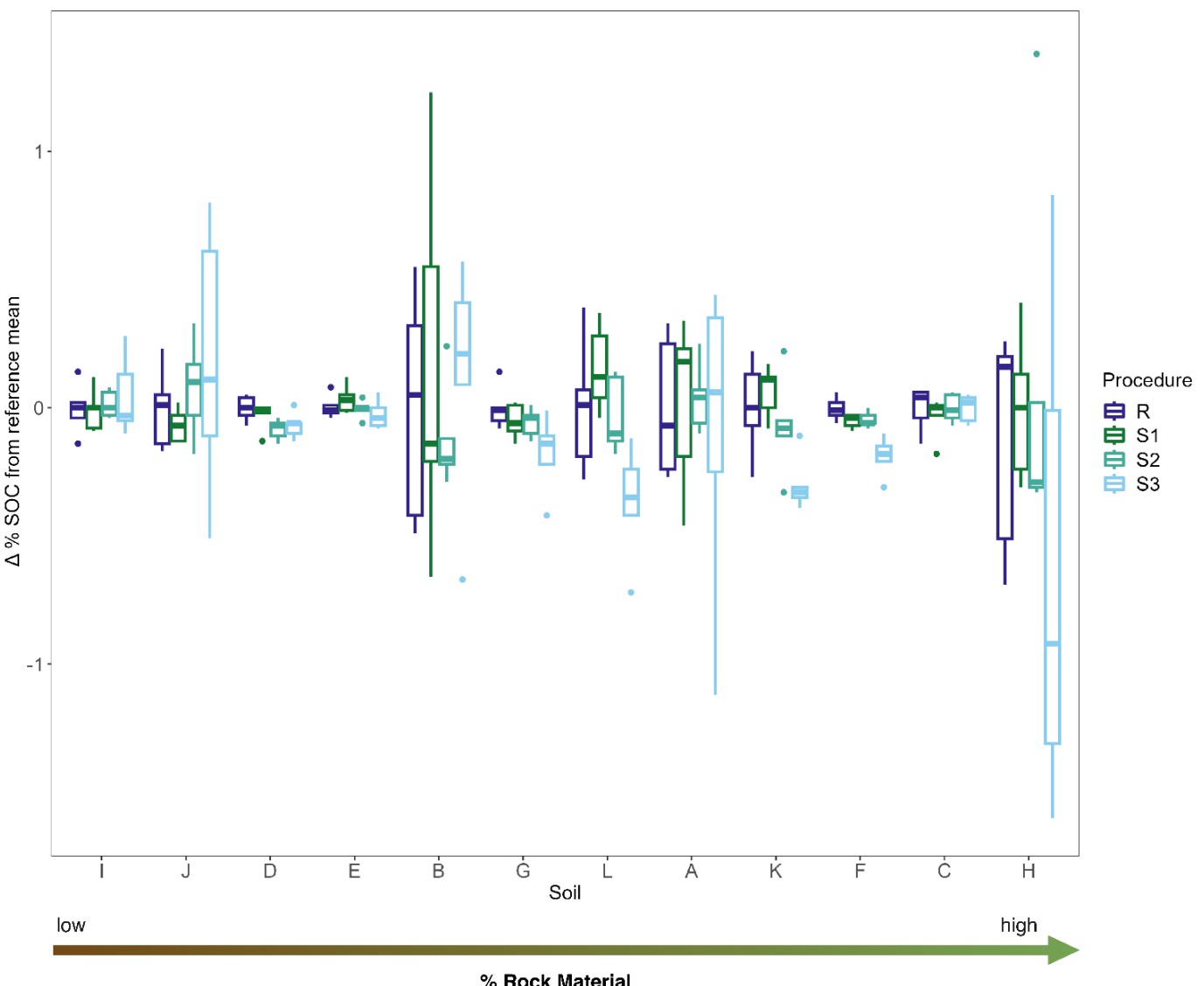

**Figure 4:** The difference (Δ) in % soil organic carbon (SOC) compared to the reference (R) mean value for all sieving procedures including R as described in Figure 1. Box plots report the median, first and third quartiles. Whiskers extend to the upper and lower data point that are within 1.5 times the interquartile range. Letters indicate the different soils, as described in Table 1, which are arranged on the x-axis by proportion of rock material removed with the R sieving procedure.

There was a significant difference in % SIC between the 8 + 2 mm (R) and 2 mm with rolling pin (S2) sieve (p=0.007), with a higher % SIC in the 2 mm with rolling pin (S2) than the 8 + 2 mm (R) (Supplemental Fig. S4 & Table S7). There was a significant main effect of sieving procedure on % SOC but no interaction with soil where we observed the

mechanical grinding procedure (S3) significantly differing from the 8 + 2 mm (p=0.003; Fig. 4), 4 mm (p=0.002), and 2 mm with rolling pin (p=0.016) sieving procedures (Supplemental Table S8). Overall, the mechanical grinding procedure was associated with a lower % SOC estimate. There were no significant differences in CV between any of the sieving procedures for % SIC or % SOC (Supplemental Fig. S3) though the CV of % SOC for the mechanical grinder was 1.4 - 1.5 times higher than that of the other sieving procedures taken together.

### 3.2.2 Effects of fine grinding on soil carbon measurements

There was a significant interaction between soil and fine grinding procedure on % TC (Supplemental Table S6). The no fine grinding procedure (G2) resulted in significantly lower % TC compared to the roller table grind (R) in one third of the soils and higher % TC in one out of twelve soils. Differences in % TC between the ball mill (G1) and no fine grind (G2) were observed in two out of twelve soils, where ball mill (G1) resulted in both higher and lower % TC. When the ball mill grind (G1) was compared to the roller table grind (R), % TC was lower in five out of twelve soils (Supplemental Fig. S5 & Table S6). The ball mill method had a lower CV for % TC than both the roller table (R; p=0.043) and no fine grind (G2; p=0.003). Fine grinding had no effect on % SIC nor CV for % SIC (Supplemental Table S7). A quarter of the soils ground on the roller table (R) had significantly higher % SOC than the ball mill (G1) and no fine grind procedures (G2). In one of twelve soils, the roller table grind (R) and no fine grind (G2) procedures resulted in lower % SOC than the ball mill grind (Supplemental Table S7). We observed differences in the CV of % SOC where the ball mill grind (G1) had a lower CV than both the roller table grind (R; p=0.006) and no fine grind (G2; p=0.001) (Fig. 5).

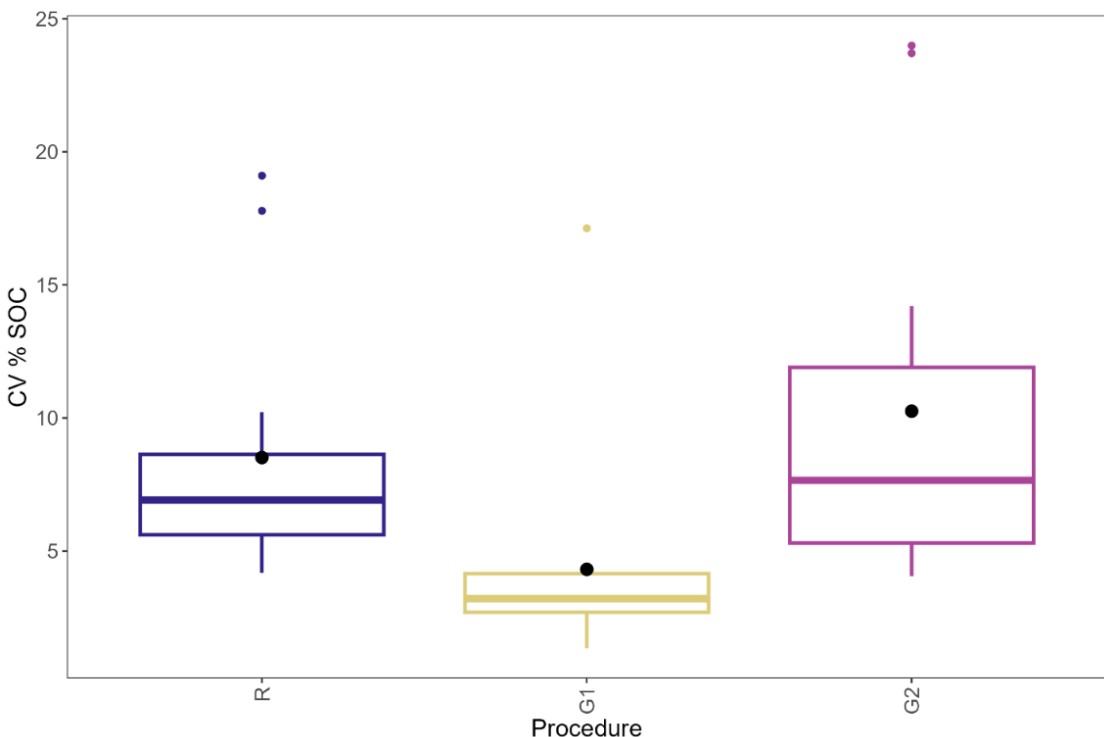

**Figure 5:** The distribution of the coefficient of variance (CV) across all soils (n=12) for each of the three grinding procedures tested, as described in Figure 1. Box plots report the median, first and third quartiles. Whiskers extend to the upper and lower data point that are within 1.5 times the interquartile range. Black dots represent the mean CV % SOC.

### 3.2.3 Effects of drying temperature on soil carbon measurements

The drying temperature prior to elemental analysis by dry combustion had a main effect on % TC, % SIC, and % SOC (Supplemental Tables S6, & S7, & S8). Air-dried soils (D2) had significantly lower % TC than soils dried at 105 °C (R; p=0.010) and 60 °C (D1; p=0.048) (Supplemental Fig. S6). Drying soils at 60 °C resulted in higher SIC than drying at 105 °C (p=0.002) and lower SIC than air-drying (p=0.039).

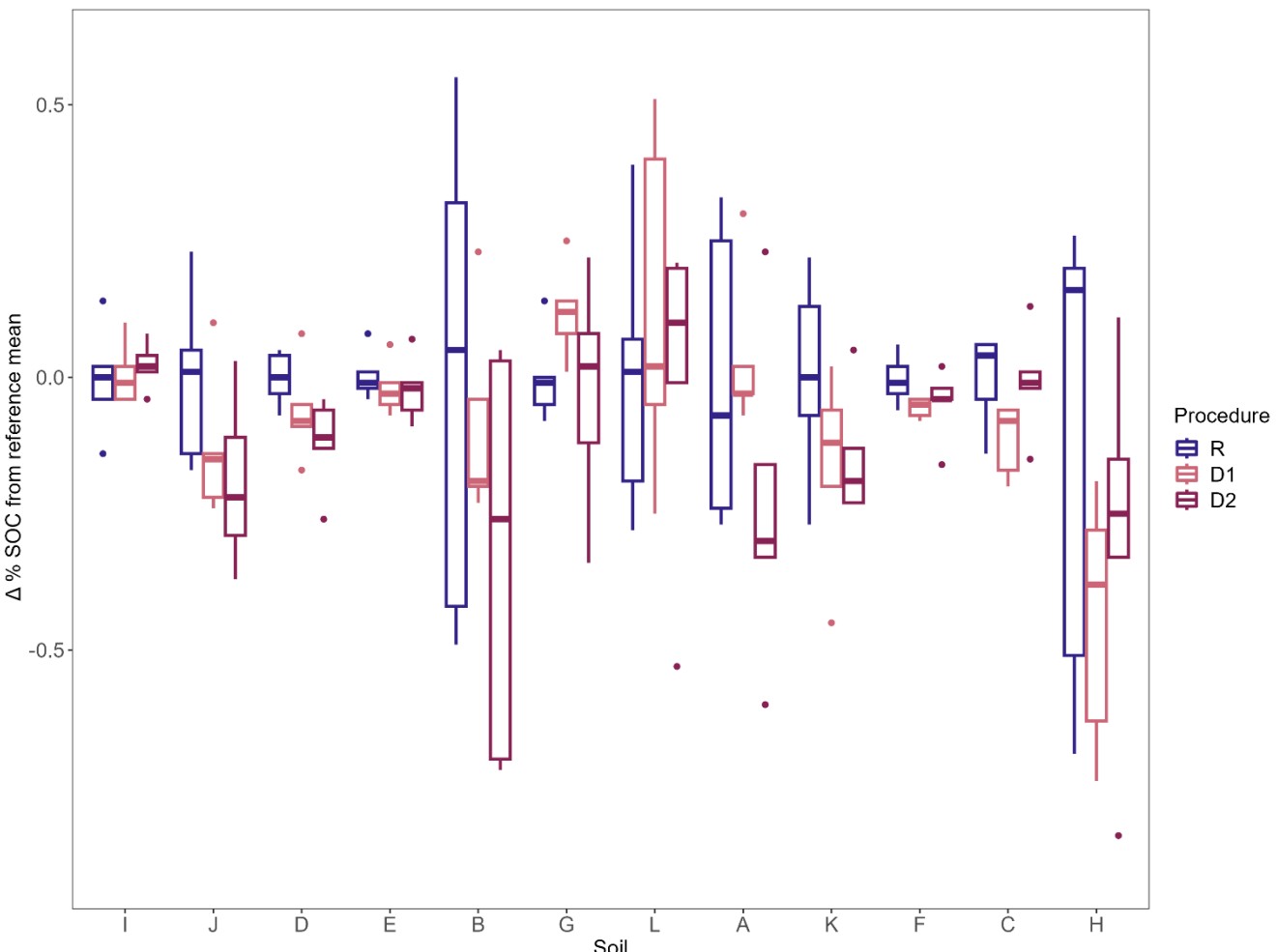

**Figure 6:** The difference (Δ) in % soil organic carbon (SOC) compared to the reference (R) mean for all drying procedures, including R, as described in Figure 1. Box plots report the median, first and third quartiles. Whiskers extend to the upper and lower data point that are within 1.5 times the interquartile range. Letters indicate the different soils, as described in Table 1, which are arranged on the x-axis by proportion of rock material removed from the R sieving procedure.

Differences in % SOC were only observed when comparing R to D2 where air-drying resulted in lower % SOC overall ($p <$ 0.001; Fig. 6). The CVs of % TC, SOC, and SIC averaged across soils were unaffected by drying temperature (Supplemental Fig. S3).

**3.3 Assessment of the effect of quantification method on soil carbon measurements**

We compared several different quantification methods to the R procedure where % C concentrations were measured by dry combustion in an elemental analyzer (EA) for % TC, by pressure transducer (PT) for % SIC, and by EA-PT for calculating % SOC. All three soil properties (% TC, % SIC, and % SOC) were predicted using FTIR spectroscopy. The FTIR
(Q1) predicted values had high linear correlations with measured values from the R method for % TC, % SIC, and % SOC. Calculating % SIC by subtracting % SOC measured by EA after acid fumigation (AF) from % TC correlated well with the PT method. The loss on ignition (LOI; Q3) method had the poorest correlation while the AF (Q2) method correlated better to the R % SOC measurements by EA-PT (or just by EA in the case of soils without SIC) for % SOC (Fig. 6). All regressions were significant at p < 0.001. Pairwise comparisons of the average CVs showed no significant differences for % TC (p=0.5214), %
SIC (p=0.217), or % SOC (p=0.18) across quantification methods.

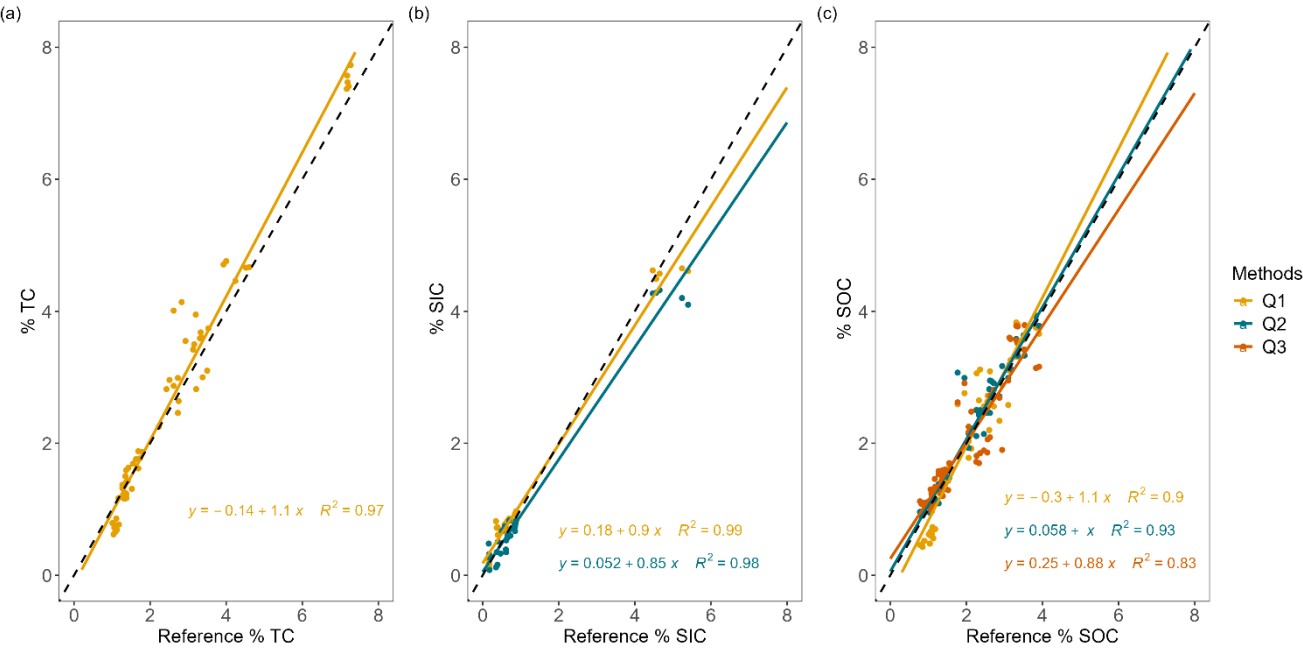

**Figure 7:** All quantification methods for % soil total carbon (TC), % soil inorganic carbon (SIC), and % soil organic carbon (SOC) plotted against the reference method where Q1 is predictions using Fourier transformed infrared spectroscopy, Q2 is acid fumigation, and Q3 is loss on ignition as described in detail in Fig. 1. The dashed line represents a 1:1 relationship.

**4 Discussion**
**4.1 High variability in soil carbon measurements provided by service laboratories**

Commercial service labs, including the labs we used for our external lab comparison, are commonly used to quantify SOC for C market applications, including demonstration projects, model validation and project verification activities (Smith et al., 2020). Given the large variation in soil C we report for the same soil, sometimes even within the same lab when two subsamples were sent, the fact that C market applications rely on these C data is very concerning. We observed the largest distribution for % SOC in the B and H soils using the CSU/reference method (Fig. 2 & 4), which both fall under the high plant material category and have SIC (Table 1). The distribution of % SOC across the external labs was also largest in soils B and H, confirming the expectation that more SIC and a larger proportion of plant and rock material will contribute to higher variability in SOC data. Within CSU, we speculate that the higher variability in % SOC in soil H was due to the measured values for % SIC using the pressure transducer since % TC variability is very low and % SOC is calculated by % TC - % SIC. We corroborate past findings that accurate C quantification is made more difficult with the presence of inorganic C, particularly at higher SIC concentrations (Stanley et al., 2023; McClelland et al., 2022). The higher variability in soil B by the CSU/reference method is most likely due to substantial amounts of fine root (irrigated pasture) and some SIC. As the interest of soil C markets extends to cropping systems with higher root biomass and grasses (Bomgardner & Erickson, 2021), for accurate C measurement it becomes paramount that plant material is being removed from soils. However, even careful sieving cannot remove all fine roots as many roots will pass through the 2 mm sieve and contribute to higher variability if not adequately removed. In soils with a high density of fine root material, extra time may be necessary to adequately remove roots using tweezers. The time could be standardized by site (soil) so that all samples receive the same treatment. It is also worth noting that a representative subsample of the whole soil could be 2 mm sieved to eliminate the time needed to remove coarse materials.

For all soils we observed a much larger distribution of soil C values across the external labs compared to across the CSU/Reference values (Fig. 2), despite the soils having been homogenized from the field the same way for subsampling. Given that the external labs use a variety of processing methods, most of which utilize a mechanical grinder and do not fine grind (Supplemental Table S2), we attribute a larger spread in % SOC across the external labs in part to the processing methods used but also due to the lack of consistency in the methods generally. Moreover, if applying the standards used by the NAPT for the acceptable range of values at +/- 2.5 times the MAD units from the median (S890 North American Proficiency Testing program oversight committee, 2020), we observed that, across all sites and laboratories (including CSU), 8 %, 15 %, and 28 % of observations were outside the acceptable range for % TC, % SIC, and % SOC, respectively. Again, this result points to differences in the soil processing methodologies affecting soil C estimates as most of the labs used in this study are NAPT certified and as previously mentioned, soils analyzed for an NAPT certification are processed in the same way.

Our findings contradict a previous inter-lab comparison study where researchers found that SOC data was agreeable between three laboratories with high correlation (Bowman et al. 2002). However, for that study, soils were sent to USDA-ARS affiliated laboratories and the samples had been pre-processed, air-dried to pass through a 2 mm sieve before shipment, finely ground by the participating laboratories before elemental analysis, and analyzed on the same type of EA (Bowman et al., 2002). We recommend that users of service laboratories at a minimum request information about the soil processing methods.

However, a more effective strategy would be to standardize soil processing methods across labs based on our suggestions below. Moreover, labs should be using a reliable instrument (e.g. have a NAPT certification) to quantify TC and SOC if using acid fumigation. In our study, it is highly likely that Lab II inadvertently swapped the sample from soil H and the air-dried sample from soil J. Since an external client would most likely not have internal analyses for comparison, we treated the data from Lab II as it was reported by the lab. Ideally, laboratories should run blind duplicate samples and include a posteriori

verification to ensure that samples are not switched for accurate reporting of the data to their clients, however given the significant added expense this would entail, improved sample tracking systems may be a more feasible solution.

## 4.2 Importance of sieving, grinding, and oven drying for reliable soil carbon measurements

### 4.2.1 Sieving

We found clear evidence that sieving soils to 2 mm after mechanical grinding of coarse materials (S3), produced different and more variable soil C data than the 8 mm + 2 mm (R), 4 mm (S1), and rolling pin + 2 mm (S2) sieving procedures (Fig. 4 and Supplemental Fig. S3; Supplemental Tables S6 & S8). Mechanical grinders are commonly used to sieve and grind soils at commercial labs (Garcia et al., 2022) because they significantly accelerate soil processing time, reducing total costs of analyses (Supplemental Fig. S7). But the speed and cost reductions come with a price of increased variability and what appears

to be lower accuracy since the data differed from the other three sieving procedures. Surprisingly, we did not observe a higher % TC or % SOC in soils with high plant material for the sieving procedures that removed less plant material. We suspect it is because soils with high plant material also had high or medium rock proportions (Table 1), with the two opposable sources (i.e., higher for plant residues and lower for rock presence) of error possibly cancelling each other out. Yet, it is not safe to assume this would always be the case, and if less rocky soils had been selected the failure to remove plant material would

likely have led to misleadingly high C measurements.

We did observe lower % TC and % SOC values using the mechanical grinder (S3) procedure in soils with either high or medium rock content (Table 1; Fig. 4; Supplemental Fig. S2). This result indicates that the presence of rock, larger than 4 mm specifically, has more impact on C data than the remainder of plant material. We speculate this is because the mass of rock (i.e., the proportion relative to the soil), is greater than that of remaining plant material. Hence, crushing large rocks

instead of sieving them out of the soil sample dilutes the concentration of TC and SOC due to the additional mass from the rock remaining. Although the higher time commitment to remove coarse materials by sieving increases cost, our results show the benefit of removing coarse material, which is that it produces more accurate and precise results. There are machines available that automate the sieving step of soil processing, but we chose not to include an automated sieving machine as one of our sieving treatments because none of the labs we surveyed use one and we have found them to be less efficient on soils

with higher clay after testing one machine internally at CSU. However, it may be worthwhile to test the effectiveness of various automated sieving machines in future studies for their potential to increase throughput. As soil sampling and analysis demand grows to meet the requirements of various policy and market-based programs, the additional time and cost may become even more prohibitive, unless there is greater value placed on standardization of lab procedures and/or program-wide requirements

to follow best practices. Increased throughput and capacity via expansion and streamlining of soil laboratories could also help to lower costs of rigorous soil processing and reliable soil C quantification.

If the goal is to obtain accurate whole soil C concentration data, our results suggest that the 2 mm sieve using a rolling pin (S2) is a sieving method that performs comparably to the 8 + 2 mm (R) method in terms of accuracy and precision but at a reduced time and cost (Supplemental Fig. S7). Although the 4 mm sieve (S1) method did not significantly differ in C data compared to the 8 + 2 mm (R) and 2 mm with rolling pin (S2) sieving methods, we do not suggest this method be included as a standard procedure because it does not remove coarse rocks from soils (Fig. 3) and there is potential impact of not removing rocks between 2-4 mm in size that we did not adequately test with our range of agricultural soils. Additionally, our study was not designed to test soil preprocessing protocols' effects on soil bulk density determination. However, accurate bulk density measurement requires the removal and quantification of rocks and coarse fragments (Rytter, 2012).

If the goal is to quantify soil C stocks using both C concentration and bulk density, we recommend the 8 + 2 mm (R) sieving method be used to ensure that sufficient coarse material is being removed to inform a more accurate soil mass. This is especially advisable in rocky soil types. However, because we observed that the 2 mm with rolling pin (S2) sieve removed as much rock material but less plant material than the 8 + 2 mm (R), we would need to scale this difference up to determine if not removing as much plant material would make enough difference in the soil mass to affect bulk density. Given that root material is so light in mass, we anticipate the 2 mm with rolling pin (S2) method could be a reliable method for bulk density measurements. Future studies should test this expectation.

### 4.2.2 Grinding

Our study demonstrated that fine grinding increases the precision of soil C measurements, as we saw on average the highest CV in the procedure where soil samples were not finely ground (G2) and a significantly lower CV for samples pulverized by ball mill to $< 125\ \mu m$ (G1; Fig. 5). The roller table grind (R), although not statistically different from the no fine grind method, did result in a lower overall CV for % TC, % SIC, and % SOC, averaging across soils, suggesting that both approaches of fine grinding over no fine grind after 2 mm (2000 $\mu m$) sieving, improve precision. This corroborates the findings of Cihacek & Jaconson (2007) who observed less variability of C data in soils ground to 150 $\mu m$ by two to six-fold when compared to soils only ground to 2000 $\mu m$. Given the high costs of ball mills and their very slow throughput, cheaper and higher throughput methods which reach a particle size of $< 250\ \mu m$, such as the roller table (Arnold and Schepers, 2004), are recommended to increase sample homogenization, thus improve precision. Given that we used approximately 10 to 30 mgs of sample for elemental analysis and Cihacek & Jaconson (2007) used around 150 mg, future research should test the effects of fine grinding using EAs that require more sample mass (i.e., 1000 mg or more), as the level of grinding may not be as important. Cihacek & Jaconson (2007) also found that with a coarser grind, medium textured soils had higher total and SOC compared to the soils ground to either 250 or 150 $\mu m$. However, our results are more similar to Dias et al. (2010) who found no differences

in C concentration data between soils ground to < 2000 μm and soils ground to < 150 μm because our results for the effect of fine grinding on C measurements are unclear.

### 4.2.3 Drying

We found no published studies that test the effect of drying temperature on C concentration data in soil using dry combustion by EA. We show that omitting oven drying leads to biased C measurements, with air-dried soils analyzed on the EA having lower % C compared to oven-dried soils, driven by the presence of moisture and its impact on sample mass (Fig. 6 & Supplemental Fig. S6). Additionally, our finding provides evidence that volatilization of SOC is not detected in soils with < 3.6 % SOC when dried at 105 °C. The potential for SOC volatilization is a valid concern but the only study we found to test

for OC volatilization prior to dry combustion corroborates our finding where there was no evidence of volatile OC loss in marine sediments dried at 110 °C and finely ground (Mills and Quinn, 1979). Additionally, we did not observe a significant effect of drying temperature on % N concentration (p=0.201; Supplemental Fig. S8). We cannot exclude the possibility that C volatilization would not occur in soils with SOC values higher than 3.6%. When analysing soils with high SOC, using a moisture correction may be preferable to oven drying the sample at 105 °C prior to EA. It is worth noting that our study was

conducted in Colorado, with a relatively low humidity of 52 % on average annually (World Data Center of Meteorology, 2024) in the Denver-metro area. Oven-drying soil samples prior to EA analyses is likely even more impactful in laboratories in more moist environments. It is well documented that texture and organic matter (OM) affect soil moisture retention (Amooh & Bonsu, 2015; Lal, 2020). Even after air-drying, soils on a texture and OM combination gradient can vary in % moisture from 0.54 - 5.22 (Wang et al., 2011). Given we did not find a statistically significant interaction of drying and soil in our model, our

results suggest that the effect of drying procedure on final SOC quantification does not vary significantly with texture or OM level. We therefore suggest that air-dried soils, generally, will result in the underestimation of % C as calculated as the mass fraction per unit of dry soil (Popleau et al., 2015). Drying had no effect on precision (CV; Supplemental Fig. S3) in our study which did not align with our hypothesis, indicating that soil moisture was homogeneously distributed across the soil mass in our soils. As we tested the effect of oven drying only on 2 mm sieved and ground soils, we cannot exclude the possibility that

higher variability on soil C values would be observed in air-dry soils which had not been carefully sieved and ground.

### 4.3 Methods for soil carbon detection

    We found a strong correlation between the FTIR predicted % TC (Q1; Fig. 6) and directly measured % TC using the EA (R). Additionally, the FTIR spectroscopy approach performed well for % SOC predictions. Others have found similar

efficacy of this approach (Zimmermann et al., 2007; Kamau-Rewe et al., 2011). For example, a recent agronomic trial reported high correlation ($R^2 = 0.99$) between FTIR-predicted SOC and SOC measured in a laboratory on all samples using mean values (Sanderman et al., 2020). The level of fine grinding needed to obtain the most accurate and precise data from FTIR spectroscopy is unclear as results have been contradictory (Wijewardane et al., 2021; Sanderman et al., 2023). However, Sanderman et al. (2023) showed that the level of grinding did not matter if the models were built from soils that were ground

to the same particle size. This observation was confirmed by our work, as we observed that grinding to < 180 µm, which is the particle size of the NRCS-KSSL spectral library (Seybold et al., 2019) we used to build our FTIR models, was sufficient to obtain reliable predictions. In our study, FTIR predictions were affected by the particle size after grinding for % SIC and % SOC, but not for % TC (Supplemental Fig. S1). The FTIR spectroscopy method may thus be a good alternative to EA as it is both reliable and more time and cost efficient. It is worth noting that we obtained accurate results for the FTIR method because

we used the same protocols and the same instrumentation for scanning our soils as was used to build the NRCS-KSSL library. Building models using samples processed differently or analysed using different instruments may have produced different results. Additionally, the KSSL library used in our study was representative of the geographical region for our sample set. The effectiveness of FTIR coupled with predictive modeling depends on the accessibility of spectral distribution within the geographical area of interest. Projects may be limited by the spectral libraries available.

We found that correlations of % SOC when compared to the R method were generally less agreeable (i.e., lower $R^2$) than for % TC or % SIC, although still high. Loss on ignition (LOI; Q3) was the least agreeable to the R % SOC, generally underestimating % SOC. However, LOI still showed high precision, with an average CV that was lower than the EA-PT method (R), FTIR (Q1), and acid fumigation (AF; Q2) (Supplemental Fig. S3). It is important to note that LOI was performed on the 8+2 mm sieved soil (R) where coarse material was diligently removed. If LOI was used to quantify SOC from soil

samples sieved using a mechanical grinder, for example, the CV may be higher. Further, we were consistent with how we employed the LOI method. The more specific deviations in methods common to LOI-based approaches can vary in important ways such as time of exposure to high heat and final temperature, which would impact the final numbers. An inter-lab comparison study published in 2001 found a "laboratory-specific pattern in the results" whereby ten labs used their own method rather than a standardized LOI method (Heiri et al., 2001).

Based on our study, we suggest that FTIR is a more precise method for SIC quantification than acid fumigation (AF; Q2). Although the % SIC calculated by FTIR (Q1) and EA-AF (Q2) correlated similarly to the pressure transducer method (R), the relative CV for FTIR was 70 % lower than that of EA-AF (Supplemental Fig. S3). These results indicate that calculating % SIC as % TC - % SOC (Q2) is not as precise as quantifying % SIC directly using either predictions via FTIR or using a pressure transducer. As mentioned above, the accuracy of the FTIR method depends on the correspondence in terms

of protocols and instrumentation between the samples analysed and those used to build the library (Safanelli et al. 2023). It is thus recommended that laboratories intending to use the FTIR method apply the same protocols used to build the library they intend to use for their prediction models. Moreover, as the use of FTIR gains traction, laboratories need to be aware that model transfer from a large spectral library (like the KSSL) may be problematic if the instrument used to analyze the soils does not match the instrument used to build the spectral library (Safanelli et al., 2023). Most importantly, it's crucial that testing for

presence of SIC is incorporated into the standard operating procedures for soil processing in all soil testing labs, and that accurate quantification of SIC is carried out where its presence is detected. By not accounting for the inorganic C in calcareous soil, labs are overestimating true % SOC.

## 4.4 Implications of our study

Our findings have important implications for how soil C data are gathered and used, including in policy and market sampling, to either calculate the number of credits directly or to inform the use of models to estimate soil C sequestration (Zelikova et al. 2021, Oldfield et al. 2022). The variation in soil C estimates based on soil mass or bulk density estimation (Raffeld et al., 2024) and lab processing procedures observed in this study may be greater than the rates of soil C accrual measured based on the implementation of certain agricultural management practices. For example, the largest variation of %

SOC measurement across all soil processing and quantification protocols was found for soil H using the mechanical grinding (S3) sieving procedure. Assuming a bulk density of 1 g cm$^{-3}$, this procedure (S3, soil H) resulted in a standard deviation ($\pm$ 15.0 Mg C ha$^{-1}$) that was 50 % of the mean average SOC stock (27.9 Mg C ha$^{-1}$). For the external lab comparison, Lab III, soil B had the largest variation in % SOC measurements. Again, assuming a bulk density of 1 g cm$^{-3}$, this lab produced a standard deviation ($\pm$ 15.4 Mg C ha$^{-1}$) that was 30 % of the mean average SOC stock (50.2 Mg C ha$^{-1}$) between the two samples measured.

The ability to detect SOC stock changes with such high error in measured data presents difficulty with average SOC sequestration rates in agriculture of 0.3 Mg C ha$^{-1}$yr$^{-1}$ (Sanderman et al 2010, Poeplau & Don 2015). Large errors associated with soil C data can also result in a gross over or underestimation of SOC. This is especially dangerous if these inaccurate data are used to calibrate models as models may then misrepresent fundamental processes and perform poorly. Moreover, large error indicates a lack of precision which can lead to a flawed assessment of change since a measurement may be more or less

biased than a measurement taken previously (e.g., 3 years prior). Not only would this cause inaccurate assessments of GHG emissions reductions or removals but could also preclude scientific understanding of what practices work to improve soil C sequestration and soil health, generally. Variation in soil C measurements based on methodological differences that are not accounted for in current protocols matters not only for agricultural producers, but also for other stakeholders who need to have confidence in soil C quantification to incentivize certain management approaches for climate mitigation. Quantifying changes

in SOC stocks is challenging for a variety of reasons, but collecting robust data starts at the ground level with how soils are collected in the field (Minasny et al., 2017) and how those soil samples are processed, as our study demonstrates. Implementation of rigorous, standardized soil processing methods across commercial service laboratories will go a long way towards more accurate and precise soil C quantification and building confidence in soil C measurement, monitoring, reporting, and verification schemes.

We suggest a set of standards and guidelines to be adopted for soil processing and quantification of SOC. Sieving is one of the most important soil processing procedures and we suggest one of two methods be implemented to reduce variation and increase accuracy: (1) hand-sieve fresh soil through an 8 mm sieve, air-dry the sample, and then sieve the whole sample or a representative subsample through a 2 mm sieve using a mortar and pestle to gently break aggregates, as described in Mosier et al. (2021) or (2) air-dry the bulk soil and then sieve soil through a 2 mm sieve using the rolling pin method as

described by the NRCS methodology (Soil Survey Staff, 2022) to gently break aggregates. Once sieved, soil should be finely ground to decrease variability across subsamples using a ball mill or roller table, which serves as an efficient alternative. Drying soils at 105 °C for 24 hours before analysis on the EA is strongly recommended to avoid underestimation of SOC

caused by the effect of moisture on sample mass. While there have been concerns reported about using 105 °C drying temperature for % N quantification, we did not observe a significant effect of drying temperature on % N concentration (p=0.201; Supplemental Fig. S8). We recommend the use of FTIR spectroscopy for predicting C in soils, with the caveats illustrated above as discussed in depth by Safanelli et al (2023). We also encourage future research to better understand if the precision of FTIR predicted SOC measurements is sufficient to detect changes in SOC at the field scale. Finally, we do not recommend LOI to measure % SOC and instead recommend the continued use of EA-PT (R), potentially benchmarking the use of FTIR spectroscopy as an additional method for SOC quantification.

Code and data availability:

The data and R code related to this article is available online at https://zenodo.org/doi/10.5281/zenodo.11223422

Supplement:

Supplemental figures and tables related to this article are available online at
https://zenodo.org/doi/10.5281/zenodo.11223422

Author contributions:

Conceptualization: MFC, RJE, MBM, JML, TJZ. Data curation: RJE. Formal analysis: RJE, MBM. Funding acquisition: TJZ, MBM, MFC. Investigation: RJE, MBM, JML, TJZ, MFC. Supervision: MFC, MBM, TJZ. Visualization: SJL. Writing (original draft preparation): RE. Writing (review and editing): MFC, MBM, JML, TJZ. All authors have read and agreed to the published version of the manuscript.

Competing interests

Rebecca Even, Megan B. Machmuller, and M. Francesca Cotrufo are cofounders of Cquester Analytics LLC, a service analytical facility which provides the analyses of soil organic matter.

Acknowledgements

The authors would like to thank Will Simescu for their assistance in the lab and Morgan Margaret for helping us bring Figure 1 to life

Financial support

This work was supported by the Environmental Defense Fund with an award from the Earth Fund.

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
