# Peer review of "Large errors in soil carbon measurements attributed to inconsistent sample processing"

_EGUsphere, 2024_

## Author Comment (AC1)

**REFEREE 2**

The manuscript submitted to 'Soil' touches a highly relevant topic, namely the correct quantification of soil organic carbon (SOC) plus other carbon species to realistically represent soil carbon (not only) in sequestration claims.

The presented work is based on some kind of round-robin analysis of aliquoted soil material which had been prepared by the authors and shipped to various laboratories for subsequent quantitative C-analysis.

While I consider the motivation and overarching idea certainly worth for a SOIL contribution, the quality of the present status of the manuscript does not permit acceptance. In the following, I go through the manuscript from beginning to end and point out the present weaknesses – regardless of whether this is a very minor issue or a bigger one.

> **RESPONSE:** We thank you for your enthusiasm surrounding the topic of our study. We address your comments point by point below in hope of improving the manuscript for publication if the editor allows.

**Abstract**

**Line 19/20**: A mechanical grinder is no instrument for sieving.

> **RESPONSE:** A variety of mechanical grinders are used for processing soils to pass through a 2 mm screen. Commercial labs often use these grinders (as we point out in our study; Supplemental Table 1) and simply report in their methods that "soils were sieved to 2 mm". Hence, we included this method within the sieving treatments, while agreeing with the reviewer that it is not a traditional sieving method. This is clearly documented in the text.

**Lines 22/23**: That finer grind leads to lower variance is nothing new and can easily be explained.

> **RESPONSE:** It's true that this is considered by soil scientists as a "known" result of finely grinding soils prior to EA, and thus fine grinding is a common practice in research labs. Yet there are few published studies to support this belief (we cite the only two we could find in the text) that can be used to mandate grinding by commercial labs, and in the absence of this evidence, we found that many commercial labs offering soil C analyses for service do not finely grind soil. We therefore hope that the evidence presented in this paper can serve to encourage commercial labs to adopt a method that achieves a finer grind, leading to less variable soil C values.

**Lines 23/25**: Not drying soil samples prior to further processes leads to errors similarly is nothing new.

> **RESPONSE:** The same rationale presented above for grinding applies here. Again, there are very few if any publications that provide evidence that not drying soils to remove residual moisture results in more variability and/or inaccurate estimates of C, and none we could find that directly quantified the impact of drying on final C quantification. We are happy to provide evidence of this from our study that can be cited in future research. Some commercial labs we spoke with dry soils beyond air-drying or apply a moisture correction, however, most that we contacted said that they simply air-dry soil samples. Again, we hope that with this published evidence clients can be more aware of potential pitfalls, and that more commercial labs are encouraged to use oven-dried soils to improve the rigor of soil C quantification by EA.

**Introduction**

**Lines 45/46**: "sample preparation is considered the first step…". This perception of the authors underlies various expressions of this manuscript, although they do refer (towards the end) to Minasny et al. (2017), where it is correctly argued that the field sampling design is by far the largest source of error. The – in my eyes – slightly distorted relevance of all subsequent steps (independent of the fact that these are relevant, too) reverberates throughout this manuscript and may lead to misperceptions with unexperienced readers and people who prefer to seek the mistakes in the laboratory works and not in their own field work.

> **RESPONSE:** We couldn't agree more that the sampling design and method used for soil collection is important, especially when trying to detect a change in SOC stocks over time, and that's why we note it in the manuscript, as recognized by the referee. However, several other papers discuss the importance of sampling design, which is a well-recognized topic, out of the scope of this study. In this study, we want to draw attention to a much less recognized yet important factor in producing accurate soil C values, and specifically test the effect of soil processing and quantification methods on % TC, SIC, and SOC estimates. However, we appreciate the reviewer's point of view so will remove the text "the first step" add text to the revised manuscript as detailed below.

> **Proposed text (L42):** Major sources of error when trying to determine changes in SOC from a baseline measurement are the sampling design and location for resampling (Rawlins et al., 2009).

**Lines 68ff**: The discussion on sieving here and later again appears somewhat odd to me. It is known that soil material must be dried (minimum air-dried, better 40°C) prior to sieving and that optimum sieving results also demand humidity control in the sieving lab to avoid badly reproducible results.

> **RESPONSE:** We respectfully disagree with the referee here and could not find a relevant reference to support these claims that sieving should occur only on air-dried soil and be performed in a humidity-controlled environment. We agree that a humidity-controlled room would be ideal for certain soil analyses but, as far as we know, this is not a common feature in soil processing rooms, especially for commercial labs. In our combined decades of experience, sieving fresh soils is perfectly acceptable, and, in fact, required if soils are being tested for microbial biomass carbon or water stable aggregates, for example.

**Lines 84ff**: After dry-sieving (2 mm), plus checking for possible remaining fine root material which will have to be removed by handpicking, the soil samples must be ground to analytical grade. The best results with the lowest standard deviation are obtained with a grain size smaller 63 micrometers. This is of particular relevance if methods like elemental analysis (EA) with very low inweights are being used (the authors refer to a machine by Elementar that is specifically designed to serve isotopic work. The standard machine, e.g., EL Cube by Elementar, takes maximum inweights of 20 to 50 mg), demanding maximum analytical sample homogeneity.

> **RESPONSE:** Thank you for pointing out the hand-picking process. We handpicked fine plant materials and will add that detail to the manuscript as detailed below.

> **Proposed text (L63):** Fine plant materials that are larger than 2 mm but still pass through a 2 mm sieve are often hand-picked using tweezers.

We are not aware of a reference we could include supporting the claim that grinding to less than 63 μm gives the best results. We agree that it is considered best practice in academic laboratories to fine grind soils to obtain higher precision in the data, however, in our experience, most commercial labs (and even academic/service labs) do not finely grind soils, especially to that small of a particle size. Even the ball mill that we used in our study is advertised as grinding soils to a particle size of < 125 μm. It's true that the EA used in our study can be paired with an Isotope Ratio Mass Spectrometer. In the manuscript we provided the masses of soil used for our instrument (L234): "The mass of soil used was related to its % TC where approximately 30 mg of sample was used for low % TC soils and 10 mg was used for soils considered to have medium % TC" and mention in the discussion that the level of fine grinding prior to EA may not be as important in EAs that require more mass per sample (L478 – 480): "Given that we used approximately 10 to 30 mgs of sample for elemental analysis and Cihacek & Jaconson (2007) used around 150 mg, future research should test the effects of fine grinding using EAs that require more sample mass (i.e., 1000 mg or more), as the level of grinding may not be as important."

**Lines 98ff**: The statement relating to neutral or basic pH soils is incorrect. Even soils with highly acidic pH (3.5 to 4.5) can show significant amounts (percentages) of inorganic as well as of organic carbon. Ferralsols/oxisols from the inner wet tropics serve as example.

> **RESPONSE:** Thank you for making this important point! We rarely work with tropical soils, so the possibility of a low pH soil with carbonates was not considered. We will add "For typical midwestern U.S. soils" to L98 in the revised manuscript.

**Line 100**: must read 'Soil Survey Staff'

> **RESPONSE:** Thank you. We will add "Staff".

**Line 111**: check year of McCarty et al (2010) in reference list

> **RESPONSE:** Great catch. Thank you, we will fix the mistake in the reference list.

**Materials and Methods**

**Line 136**: I suggest splitting the very long table caption into a concise header and to move the details into a table footer to make the table more appealing. Instead of 'soil identification **number**', it should read e.g., 'code' since no numbers are being used. The sequence of the table column headers should be repeated in the table header – no different sequence.

> **RESPONSE:** Thank you for these suggestions to improve Table 1. We agree that the caption is very long. We will adjust the table and caption as suggested.

**Line 145ff**: The initially stated criticism on the authors bias with the lab parts emerges here once again. To take one single sample of a 50x50 cm x 15 cm deep soul pit is radically insufficient to represent, e.g., a hectare. I suggest to simply rephrase the experimental setup from the onset and clearly and unmistakably explain that while the biggest mistakes occur in inappropriate sampling, this paper focuses on all subsequent steps and uses homogenized soil samples to test sample preparation and analysis steps.

> **RESPONSE:** We apologize if it appears we overemphasized the relative importance of lab protocol versus field sampling. As stated above we do not intend to claim that it is "more"

impactful on final quantification, rather that it represents another critical factor affecting final SOC values that requires attention (alongside field sampling which is addressed in other studies). We appreciate the concern for readers. We want to make sure readers understand that we are not principally concerned with obtaining a representative field sample from the sites we visited. To answer our research questions, we needed the soil to be as uniform as possible for each procedural variation and replicate so that spatial heterogeneity was not a major factor driving our results. We will revise the sentence as detailed below:

**Current text (L145):** Soils were collected by spade from roughly a 50 cm x 50 cm area

**Proposed text:** To collect a relatively uniform sample and avoid a strong influence of spatial heterogeneity, soils were collected by spade from a small area, roughly 50 cm x 50 cm. The intention of this sampling procedure was not to obtain a sample that represented the field site or a large area (e.g. on the hectare scale), rather only to collect enough soil with unique (relative to other sites) and uniform properties (within the collected soil) to use for the laboratory procedure comparison.

**Line 149**: 'Soil was collected from different places on the butcher paper…'. A) What is butcher paper made of? Does it contain any carbon like all other papers? If so, discuss. B) The sub-sampling description here does not suffice to allow others to judge the procedure. We generally use multi-step quartering or mechanical sample dividers to obtain true aliquots.

> **RESPONSE:** The paper we use is found here. We refer to it as butcher but will change it to "kraft" in all occurrences throughout manuscript (L148, L149, & L189). This type of kraft paper has no water-soluble carbon and is machine made from resinous wood and non-wood sources (link). We will address this and Point B in our response below.

**Line 166**: I read that the soil sample was homogenized as field moist material. This would certainly introduce possible errors since even smaller differences in soil humidity make homogenizing differ between samples or different humidities.

> **RESPONSE:** Yes, the soil was homogenized field moist since one of our sieving treatments involved sieving the soil fresh. Relevant to point B above, we homogenized to the best of our ability and collected samples from various places while it was laid out to try and minimize heterogeneity across subsamples. We included replicate as a random effect in our mixed linear models to account for this since soils were divided into their replicates first and then divided for each procedural variation. To address this comment, we will revise the text as detailed below.

> **Current text (L147):** Each field moist soil was homogenized by removing the entire sample from each bucket, spreading the whole sample out on butcher paper, and flipping the soil over itself twice prior to collection. Soil was then collected from different places on the butcher paper to ensure representative subsamples

> **Proposed text:** Each field moist soil was homogenized to the best of our ability. While it is impossible to eliminate variability due to heterogeneity across subsamples and replicates, we sought to minimize its contribution by spreading the entire sample out on kraft paper, flipping the soil over itself twice, and collecting soil from various parts of the kraft paper to ensure representative subsamples. The kraft paper used has no water-soluble carbon. Further, we subsampled for all replicates, including those sent to external laboratories in the same way to

minimize differences across laboratories (e.g. between CSU and external labs, see Fig. 2) due solely to soil heterogeneity.

**Line 169ff**: all SI units must be set with a space between number and unit. This is valid throughout and should be corrected in the entire manuscript.

**RESPONSE:** Thank you for pointing this out. We will add a space here and throughout the manuscript as suggested.

**Line 180ff**: Similar to Table 1, the header should be split into a concise header and detailed explanation below the table as a footer. The table itself prints badly in my copy. Please check.

**RESPONSE:** We agree that this table needs improvement. We will create a figure to replace this table using the appropriate headers and a more concise caption. We propose the figure below to replace this table in the revised manuscript.

[Figure]

**Figure 1**: The procedural variations for sieving, grinding, drying, and quantification methods of total carbon (TC), soil inorganic carbon (SIC), and soil organic carbon (SOC) concentrations. Sieving variations include the Reference (R; 8 + 2 mm), S1 (4 mm), S2 (2 mm with rolling pin), and S3 (mechanical grinder). Grinding (G) variations include R (roller table grind to < 250 μm), G1 (ball mill to < 125 μm), and G2 (no grind; < 2000 μm). Drying (D) variations include R (105 °C), D1 (60 °C), and D2 (air-dried only). For the quantification (Q) of TC, dry combustion by elemental analyzer (R; EA) and Fourier transformed infrared spectroscopy (Q1; FTIR) were tested. Quantification for SIC was tested using a pressure transducer (R; PT), FTIR (Q1), and acid fumigation (Q2; AF) where SIC is calculated by subtracting TC (EA with no AF) from SOC (EA post AF). SOC quantification procedures included subtracting SIC (PT) from TC (EA) concentrations (R), FTIR (Q1), AF (Q2), and loss on ignition (Q3; LOI).

**Line 230**: Instrumentation nomenclature needs to be homogenized throughout (compare with line 243).

**RESPONSE:** We will reformat the introduction of the FT-IR used to read (L243): "…on a Bruker VERTEX 70/HTS-XT INVENIO-R FT-IR (USA).

**Line 231**: To reduce possible misunderstandings, introduce a comma between '% TC' and 'and % SOC'…

**RESPONSE:** Respectfully, we would prefer not to add a comma as we do not think one is needed.

**Line 237**: a unit is missing after '0.04'. Personal remark: Our lab regularly obtains a lower limit of determination of 0.04 wt-% in standard application for C and a related SD between 0.02 and 0.04 wt-% on an EL Cube).

**RESPONSE:** We will add % TC after 0.04.

**Line 251**: Check publication year for R Core Team in reference list.

**RESPONSE:** We will check. Thank you.

**Results**

**Page 10, Figure 1**. Any figure or table should never directly follow a chapter or section header. The three sub-figures display three different scales (Y-axis). That is certainly not ideal to allow for an unbiased understanding of the figure's message.

**RESPONSE:** Thanks for pointing this out. We will move the figure and make several changes based on your comments and the comments from Referee 1 as detailed below.

[Figure]

**Figure 2:** The distribution of total carbon (TC; panel a) soil inorganic carbon (SIC; panel b) and soil organic carbon (SOC; panel c) concentrations from eight service soil testing laboratories and Colorado State University (CSU). Box plots report the median, first and third quartiles for values from all soils (field moist and air-dried) analyzed at service soil testing laboratories (brown boxplot) and CSU (grey boxplot; n=5). Whiskers extend to the upper and lower data point that are within 1.5 times the interquartile range. For soils B, C, D, and J, two samples were sent to each external lab, one air-dried and one field moist (n=16). One sample from soil H was sent to each lab (n=8). Refer to Table 1 for a description of the soils, Figure 1 for Reference (CSU) methods, and Supplemental Table S2 for external service soil testing laboratory methods.

**Line 271**: Unit is missing after 'and 1.45'

    **RESPONSE:** We apologize that the unit is unclear. We will revise the text as detailed below.

    **Current Text (L270):** Within a given lab, reported values for % TC, % SIC, and % SOC for the same soil (sent as either air-dried or field moist) varied by up to 4.62, 4.06, and 1.45 respectively.

**Proposed text:** Within a given lab, reported values for the same soil (sent as either air-dried or field moist) varied by up to 4.62 % TC, 4.06 % SIC, and 1.45 % SOC.

**Page 12, Figure 2**: The procedures (x-axis) should display horizontal indicators (here P0, P1, etc.). To simplify, and since the term 'Procedure' is printed below, the number would suffice. Again, to avoid perception bias, the legend should explicitly point out that the is factor 10 between the y-axis of a) and b).

**RESPONSE:** We will change the figure (Fig. 3 in revised manuscript) using the new naming scheme we describe above in the proposed Figure 1 for the sieving treatments. The difference in the y-axis scale is pointed out in the caption but we will make the caption more concise and concise as proposed below. X-axis labels are angled for fit.

**NEW FIGURE:**

[Figure]

**Figure 3:** A stacked bar graph illustrating the proportion of coarse material removed from the total soil mass with four different sieving procedures: R (8 + 2 mm), S1 (4 mm), S2 (2 mm with

rolling pin), and S3 (mechanical grinder) described in Figure 1. Stacked bars represent the mean (± standard error; n=5) of coarse material identified as plant (top; green) or rock (base; beige). Letters refer to soils as described in Table 1. Panel a (top) includes soils with less coarse material (up to 1% on average), and Panel b (bottom) includes soils with more than 1% coarse material

**Page 13, Figure 3**: This figure prints badly. The symbols need to be bigger, and the axis formatting with black lettering and slightly larger and horizontal (x-axis) lettering.

**RESPONSE:** We will remove Figure 3 in the current manuscript based on an observation and comment made by Referee 1 which we have addressed in our response to them.

**Page 16, Figure 5**: Same as with fig. 2, including homogenized axis scales

**RESPONSE:** Fig. 2 will be moved into supplemental and replaced with a figure illustrating CV % SOC just for the grinding treatment comparisons (R replaces P0, G1 replaces P4, and G2 replace P5 in the revised manuscript) as proposed below:

**NEW FIGURE:**

[Figure]

**Figure 4:** The distribution of the coefficient of variance (CV) across all soils (n=12) for each of the three grinding procedures tested, as described in Figure 1. Box plots report the median, first and third quartiles. Whiskers extend to the upper and lower data point that are within 1.5 times the interquartile range. Black dots represent the mean CV % SOC.

**Page 17, Figure 6**: While again the axis scales should be equal, this figure is somewhat odd to me and appears to compare "apples and pears". Direct comparison is only possible with one modification of degrees of freedom.

**RESPONSE:** The panels are not meant to be compared to each other and the y-axis scales are different because each panel has a different variable on the y-axis. We show here how the various quantification methods we tested correlate to the measured, reference values for each variable. We believe the way in which we present these data and provide the linear regression equation and p-value is acceptable.

**Discussion**

**Lines around 421**: I cannot agree with these conclusions/recommendations. It should go without saying that only experienced laboratories that adhere to GLP do qualify. That implicitly means that there is a very tight quality control and documentation. No other labs should be considered. To determine organic carbon (TOC), acid fumigation is a necessity. However, the related process must be clearly defined.

**RESPONSE:** We respect the referee's opinion to disagree with our recommendation but wonder if our recommendation here was unclear. The NAPT certification we refer to in L421 was introduced earlier in L53. Because NAPT certified labs are sent soils that have been processed in the same way, the certification indicates that whatever instrument they are using to procure soil C estimates is accurate. And while some certifications may cover other laboratory procedures, they do not necessarily guarantee that a given lab adheres to them in all cases, and shortcuts may be taken without a client's knowledge. Further, we based our study on direct communication with several popular commercial laboratories that are regularly used by clients seeking to quantify SOC stocks and changes for real projects. Regardless of whether "no other labs should be considered", our conclusions and recommendations are relevant because these labs are being actively used. There are also commercial and service labs that are not NAPT (or alternative) certified so the client would have no way of knowing if the data is "in the ballpark" or not. We recommend the client use labs that have an NAPT certification (or comparable). That seems in line with what this comment suggests. Organic carbon is determined using methods that do not involve acid fumigation, as we have included, explained, and tested in our study. We define the other methods as being LOI using a conversion factor of 0.58 of % SOM to % SOC or calculating % SOC by the difference of % SIC from % TC if inorganic C was measured. In soils without SIC, SOC is equal to TC and acid fumigation does not need to be performed. We also include predicting % SOC using FTIR spectroscopy as a promising method.

**Lines 453ff**: Here and at other occasions, the authors point out lab costs for some more time-consuming procedures. I like to remind them that the by far most costly part of obtaining decent analytical results for anything is high-quality sample acquisition. The rest is relatively cheap and should not serve to argue for cost-savings. More precisely here: 2 mm sieving should be beyond discussion. One my sieve 8 mm or whatever in the field already to reduce the material to be transported to the lab and kept on hold in freezer or fridge, but that is irrelevant in this context. The authors seem not to know automated sieving machines (e.g., Fritsch, Retsch) that allow lab personnel to do other work while the sample(s) is being sieved. Automated sieving comes with the added advantage that it increases reproducibility of the process.

**RESPONSE:** The referee is correct in pointing out that we do not mention automated sieving machines in the current manuscript. Thank you for bringing that to our attention. The authors have one in their lab and have used it before. However, in our experience it only produces good results in sandy soil with low aggregation, and otherwise aggregates larger than the sieve mesh do not pass through, making the process inefficient. That, combined with the fact that none of the labs we interviewed used it, is why we did not include this procedure in our test. We will address this lack of recognition in the revised manuscript in the discussion as detailed below. Depending on the research questions, we do not recommend sieving fresh soils in the field unless you have a

field scale with you to weigh the samples field moist to apply a moisture correction for an accurate bulk density value later. We disagree that the time to process soil should not be considered in our discussion. The turnaround time expected in commercial labs is astoundingly fast, as clients want their data as soon as possible. Further, cost savings are imperative for commercial labs that often operate on thin margins and so can be a major factor in deciding commercial lab protocols.

**Proposed text (L448):** There are machines available that automate the sieving step of soil processing, but we chose not to include an automated sieving machine as one of our sieving treatments because none of the labs we surveyed use one and we have found them to be less efficient on soils with higher clay. However, it may be worthwhile to test the effectiveness of various automated sieving machines in future studies for their potential to increase throughput.

**Line 460**: Check spelling for Ryterr 2012

**RESPONSE:** Thanks. We will fix the spelling.

**Line 461**: In consequence to what I expressed above, I cannot agree with the suggestion made here.

**RESPONSE:** We are sorry the reviewer disagrees with our suggestions. We hope that all the clarifications provided above will help readers to better appreciate our points. Our suggestions are based on the findings of our research. We certainly could have included more treatments and/or used soils on a larger gradient of % SOC, texture, etc. However, we had to limit our treatments based on feasibility, time, and affordability. Moreover, we included processing and quantification methods that are commonly used in U.S. commercial soil testing labs based on direct communication with popular labs. We believe we did our due diligence to create a robust experimental design and use robust statistical analyses to support our conclusions and recommendations.

**Line 470**: Grinding just like sieving should be free of individual bias. There are various mills on the market that allow for multiple (up to 8) samples to be ground to analytical grade in a few minutes with almost perfect homogeneity (as shown by laser granulometry).

**RESPONSE:** We do recommend using a ball mill to achieve the highest precision and do not doubt that some commercial labs use the grinders the referee points out. We hope that more labs adopt a fine grinding method after seeing the results of our study. If there's a ball mill on the market with high throughput, that would be ideal.

**Line 488ff**: Again in addition to what I wrote above on drying, air-drying (20–25 °C) is the conditio-sine-qua-non. Yet, if no other critical analyses (e.g. mercury) need to be undertaken on that material, then 40–60 °C drying is better since it compensates for inhomogeneities in laboratory climatology. See also line 491.

**RESPONSE:** We agree that drying beyond room temperature is better for elemental analysis, as we have presented evidence for in this study. However, we do neglect to include the option of applying a moisture correction instead of drying the whole sample for EA analysis. We will be sure to include this as a potential option in the discussion section as detailed below.

**Proposed text (L565):** However, an alternative, especially if soils with high % OC are being analyzed, is to include a moisture correction so that the true oven-dried soil mass is being input into calculations for % C determination.

**Line 494ff**: I do not understand the argumentation that their 'results were not texture or OM-dependent…' How so?

**RESPONSE:** We are sorry this was not clear. We will revise the text as detailed below.

**Current text (L495):** Our results were not texture or OM dependent since there was no interaction of drying and soil, so suggest that air-dried soils, generally, will result in the underestimation of % C as calculated as the mass fraction per unit of dry soil (Popleau et al., 2015).

**Proposed text:** Given we did not find a statistically significant interaction of drying and soil in our model, our results suggest that the effect of drying procedure on final SOC quantification does not vary significantly with texture or OM level. We therefore suggest that air-dried soils, generally, will result in the underestimation of % C as calculated as the mass fraction per unit of dry soil (Popleau et al., 2015).

**Line 516**: better to use 'it is'

**RESPONSE:** Good catch. We will change it.

**Line 525**f: Direct comparisons are only possible within one methodologically-consistent approach. One can run the EA prior to sample acidification to obtain TC, then run another aliquot after acidification to obtain TOC – the difference of which allows for the calculation of TIC. To shift instruments (methods) and determine, e.g., TC with one technique (e.g., Leco CS-analyzer), then TOC with EA is no good idea to obtain high-quality results. However, if done correctly in all steps, then you must expect very small errors between TC and TOC results from one and another method.

**RESPONSE:** We agree with the referee that it is ideal to vary as little as possible between methods for direct comparisons to determine whether a given step has an impact. The first way of obtaining % SIC that you described is what we did for our method abbreviated as P9 (which will be changed to Q2 in the revised manuscript). We correlated that to our reference method (R which is P0 in the current manuscript). We agree that it's important to use the same instrument when analyzing soils for % TC and then analyzing them again after acid fumigation for % SOC. We used the same instrument in this study and will add text to the manuscript to make this suggestion clear.

**References**

The reference list demands homogenization in formatting, bibliographical completeness, and accuracy of all citations. See, e.g., Bates et al. 2015; Bernoux and Cerri 2005; Lenth 2022; McCarthy et al. n.d.; R Core Team 2022.

**RESPONSE:** Thank you for being so thorough and checking our reference list. We will make the necessary changes to make all reference formatting consistent and correct any mistakes.

**Bottom line**: As already mentioned, the motivation of the authors deserves applause. However, the submitted manuscript falls somewhat short to deliver what it takes in order to meet the self-set goals. I suggest a thorough revision prior to re-submission.

> **RESPONSE:** Thank you for your time reviewing our manuscript. If the editor allows, our manuscript will be improved with your edits and comments in mind. We hope that our answers to your comments above have made the objective of our study clearer.

---

## Author Comment (AC2)

**REFEREE 1**

In their manuscript the authors present the uncertainty of total carbon, soil inorganic carbon and soil organic carbon measurements depending on sample processing and measurement. The authors show substantial differences that are mainly driven by sieving and measuring methods with LOI being highly variable. It is or great importance to have such comparisons and critical assessments. The need for accurate soil C measurements is getting more important for an evolving C market. A substantial overestimation of C changes would be bad for the actual climate effect and a substantial underestimation of soil C would reduce the economic benefit of the C market. The experimental set up using 11 procedures and the comparison with 8 commercial laboratories is an important approach to reach a better homogenization of analyses approaches and make soil C measurements more consistent.

> **RESPONSE:** Thank you for your time and thorough comments. We appreciate your recognition of the merits of our study. Your feedback is very valuable and we hope that by addressing your comments, we have improved the manuscript for publication if the editor allows.

I have two main concerns:

The authors need to elaborate their discussion on the application of chemometric approaches by combining MIR and predictive modelling (e.g. Line 505-512, 523-526 and 567-570) It is true that such approaches can work well as reported in the cited literature. However, it needs to be clear that this all depends on the availability of a representative soil spectral library that it large enough to develop models for prediction. The good prediction in this study is expected and bias the generalized conclusion. The model was trained on the KSSL and thus covers the spectral variability of the soils used here. Additionally, the sample pre-treatment was very similar between the P0 method here and the initial data for the model presented in Seybold et al (2019). Seybold et al (2019) measured TC by dry combustion, used the pressure transducer method for SIC and determined SOC by difference. Thus, it is a good model for the soils selected here. However, the transferability of such models is difficult and a major challenge to overcome. For example, sample grinding is important for the transferability. Grinding was also an aspect that motivated the authors to test. It is true, the differences in grinding are not so significant when all samples are similarly prepared and the model trained for the corresponding grinding is applied but transferring a model trained on finely milled sample to samples that are coarser and vice versa brings uncertainty and challenges (Sandermann et al., 2023). Recently, Safanelli et al. (2023) reported that even combining spectra obtained from different devices can be difficult and requires important pre-processing. More importantly, the authors report that sample processing resulted in larger uncertainty of the predictions. Therefore, the authors need to constrain their conclusion here that such approaches are only working when the conditions of a good and regional model are given. Otherwise, the model error (e.g. RMSE) will be too large to detect changes in TC, SOC and SIC.

> **RESPONSE:** Thank you for turning our attention to some of the reasons why we obtained such accurate results from the FTIR analyses. We agree with these concerns and are sorry for having omitted discussing these limitations in the current manuscript. We will extensively and clearly present limitations in the revised manuscript, both in the methods and the discussion of results, as most appropriate. Specifically, we will present the advantage of having the KSSL spectral library available and reiterate the importance of having a robust library of samples derived from the same region of study, built on samples pre-processed with similar approaches, and scanned with the same protocol/instrument when using the FTIR approach to estimate soil properties. We will also refer to Safanelli et al (2023) to point to the difficulties associated with FTIR predictions when these conditions are not met.

We will revise the manuscript text as detailed below:

**Current text (L23)**: The test suggested that the < 180 μm grind was sufficient for FTIR scanning and we used that for the comparison of P8 to the other quantification methods

**Proposed text:** The test suggested that the < 180 μm grind was sufficient for FTIR scanning, which was also the particle size of the samples used to build the NRCS-KSSL spectral library (Seybold et al., 2019) used in this study. Thus, we compared the Q1 < 180 μm protocol to the other quantification methods."

**\*P8 becomes Q1 in the revised manuscript as detailed below in response to the next comment**

**Current text (509):** The level of fine grinding needed to obtain the most accurate and precise data from FTIR spectroscopy is unclear as results have been contradictory (Wijewardane et al., 2021; Sanderman et al., 2023). In our study, FTIR predictions were affected by the particle size after grinding for % SIC and % SOC, but not for % TC (Supplemental Fig. S1). The FTIR spectroscopy method may thus be a good alternative to EA as it is both reliable and more time and cost efficient.

**Proposed text:** The level of fine grinding needed to obtain the most accurate and precise data from FTIR spectroscopy is unclear as results have been contradictory (Wijewardane et al., 2021; Sanderman et al., 2023). However, Sanderman et al. (2023) showed that the level of grinding did not matter if the models were built from soils that were ground to the same particle size. This observation was confirmed by our work, as we observed that grinding to < 180 μm, which is the particle size of the NRCS-KSSL spectral library (Seybold et al., 2019) we used to build our FTIR models, was sufficient to obtain reliable predictions. In our study, FTIR predictions were affected by the particle size after grinding for % SIC and % SOC, but not for % TC (Supplemental Fig. S1). The FTIR spectroscopy method may thus be a good alternative to EA as it is both reliable and more time and cost efficient. It is worth noting that we obtained accurate results for the FTIR method because we used the same protocols and the same instrumentation for scanning our soils as was used to build the NRCS-KSSL library. Building models using samples processed differently or analysed using different instruments may have produced different results.

**Current text (L525):** These results indicate that calculating % SIC as % TC - % SOC (P9) is not as precise as quantifying % SIC directly using either predictions via FTIR or using a pressure transducer. Most importantly, it's crucial that testing for presence of SIC is incorporated into the standard operating procedures for soil processing in all soil testing labs, and that accurate quantification of SIC is carried out where its presence is detected. By not quantifying the inorganic C in calcareous soil, labs are overestimating true % SOC.

**Proposed text**: These results indicate that calculating % SIC as % TC - % SOC (Q2) is not as precise as quantifying % SIC directly using either predictions via FTIR or using a pressure transducer. As mentioned above, the accuracy of the FTIR method depends on the correspondence in terms of protocols and instrumentation between the samples analysed and those used to build the library (Safanelli et al. 2023). It is thus recommended that laboratories intending to use the FTIR method apply the same protocols used to build the library they intend to use for their prediction models.

**\*P9 becomes Q2 in the revised manuscript as detailed below in response to the next comment.**

**Current text (L567):** We recommend the use of FTIR spectroscopy, particularly for SIC quantification as this method performed better than acid fumigation. Finally, we do not recommend LOI to measure % SOC and instead recommend the continued use of EA-PT (P0), potentially benchmarking the use of FTIR spectroscopy as an additional method for SOC quantification.

**Proposed test:** We recommend the use of FTIR spectroscopy, with the caveats illustrated above and those discussed by Safanelli et al (2023), particularly for SIC quantification as this method performed better than acid fumigation.

As far as I understand P0 is the reference method here but also the method used in the authors research lab. It is not clear why the authors are so certain that this method is the most rigorous. For example, in Line 161-163 the authors just argue with their "expert opinion". Many labs use ball mills that are more efficient in grinding (e.g. <50um), oven drying at 105°C might cause losses of OC in some high C soils (this is only briefly touched at the end) and the pressure transducer methods requires the direct addition of acid to the soil, which can alter the organic matter (fumigation is less harsh). The authors need a reference method here to compare to but they also need to critically discuss the constrains of P0 here. It is even more important to have a good justification here given the conflict of interest that exists here between the research and the commercial lab the authors are part of at the same time.

> **RESPONSE:** We understand these concerns and so will take a new approach for this method by simply referring to it as the reference method (R) instead of P0 in the revised manuscript. Thank you for turning our attention to the fact that we should not have used the term "most rigorous" when referring to the P0. However, we will still provide references as to why we use this method at CSU throughout the manuscript. This comment also prompted us to change nomenclature of our protocol to improve clarity. The P0 method will hereby be referred to as the reference (R) and the procedural variations will be lettered according to the processing step or quantification method being tested. P1-P3 will become S1-S3 for sieving, P4 & P5 will be G1 & G2 for grinding, P6 & P7 will be D1 & D2 for drying, and P8-P10 will be Q1-Q3 for quantification. We hope this will clear up any confusion for future readers. The proposed figure for our procedural variations is included in our comments to Referee 2.
>
> We agree that ball milling generates a more homogenized, finer sample, and thus reduces analytical error, as also shown in this study. However, ball mills are typically expensive and, more importantly, have a very low throughput, making them unappealing to commercial labs. We would be curious to learn more about a high throughput ball mill if it's on the market. It is true that OC volatilization can occur at high temperatures in soils that have high OC content, but that is rarely the case in agricultural soils, and even less in those targeted for C markets. We present evidence that C volatilization did not occur in our study drying soils at 105 ºC using soils spanning a typical % SOC range found in agricultural soils. A fair point was made about acid fumigation being less harsh on the sample. However, the pressure transducer method is destructive. The sample is disposed of and does not undergo further analyses, so those transformations have no consequences. It is also worth noting that acid fumigation is often not effective at high IC levels (typically observed in deep calcareous soils) and has been reported to affect the % N, often requested with the % SOC analysis. We chose to use the pressure transducer for the reference in this study with the combination of accuracy and efficiency in mind. The acid fumigation method is more time-consuming and expensive. For standardizing methods across labs, throughput and cost is an important consideration. The authors disclosed their relationship to Cquester Analytics and, specifically to avoid any conflict which had been discussed at length

also with the funding agency of this study, we did not involve Cquester Analytics in any of this research. Throughout the manuscript, we discuss the pros and cons of all methods, and users can make up their mind as to what is most appropriate to fit their needs.

We will revise the manuscript as detailed below:

**Current text (L161):** Each protocol, labelled as P0-P10 (Procedure 0-10), was replicated five times per soil for all 12 soils. To our expert opinion, P0 included the most rigorous procedure at each step and all other protocols deviated from P0 for one step to enable the evaluation of the effect of each individual step on the estimation of TC, SIC and SOC concentrations.

**Proposed text:** Each protocol was replicated five times per soil for all 12 soils. We considered the methods used in the Soil Innovation Lab at Colorado State University (CSU) as the reference (R) where all protocols deviated from R for one step to enable the evaluation of the effect of each individual step on the estimation of TC, SIC and SOC concentrations.

**Proposed text (L216):** The dry combustion method (R; EA) is considered the most accurate method for total C quantification (Yeomans & Bremner, 2008) so it is often used as a reference (Leong & Tanner, 1999; Bisutti et al., 2004) against other quantification methods. SIC concentration was determined using the pressure transducer as the R method because, in our experience, it is a more efficient and cost-effective way to quantify SIC compared to acid fumigation (TC – SOC) where soil samples must be analyzed twice on the EA.

**Proposed text (L490):** Additionally, our finding provides evidence that volatilization of SOC is not detected in soils with < 3.6 % SOC when dried at 105 °C. The potential for SOC volatilization is a valid concern but the only study we found to test for OC volatilization prior to dry combustion corroborates our finding where there was no evidence of volatile OC loss in marine sediments dried at 110 °C and finely ground (Mills and Quinn, 1979). Additionally, we did not observe a significant effect of drying temperature on % N concentration (p=0.201; Supplemental Fig. S9). We cannot exclude the possibility that higher variability in soil C measurements would be observed in air-dry soils which had not been carefully sieved and ground, or that C volatilization would not occur in soils with higher SOC. If analysing soils with higher SOC, using a moisture correction may be preferable to oven drying the sample at 105 °C prior to EA.

**Specific comments:**

Line 14: Please specify what "involvement in SOC quantification for C markets" means

**RESPONSE:** We will specify what we mean by "involvement in SOC quantification for C markets by proposing the text (L14) read "involvement in SOC data curation used to inform C market exchanges, which could include demonstration projects, model validation and project verification activities."

The abstract contains many details but no conclusion of the study.

**RESPONSE:** Thank you for pointing this out. We will add a sentence to the end of the abstract describing the conclusions as detailed below.

**Proposed text (L30):** We suggest that sieving to < 2 mm with a mortar & pestle or rolling pin to remove coarse materials, drying soils at 105 °C, and fine grinding soils prior to elemental analysis will improve accuracy and precision of soil C measurements. Moreover, we show promising results using FTIR spectroscopy coupled with predictive modeling for estimating % TC, % SIC, and % SOC.

Line 53: Please specify if the authors mean "quality assurance and quality control"

    **RESPONSE:** Yes, and we will specify this is the manuscript.

Line 54: Please specify NAPT for readers that are not familiar with US organisation. This holds true for all other abbreviations that are not explained.

    **RESPONSE:** Great suggestion. We will add more information about the North American Proficiency Testing program, and we'll go carefully through the manuscript to define all the abbreviations for readers. Specifically, we plan to add the following text.

    **Current text (L53):** Soil testing labs can elect to participate in QA/QC certification programs that promote their data as high quality. For example, the North American Proficiency Testing (NAPT) Program is offered in the U.S., with over 130 NAPT certified soil and/or plant testing facilities. To gain certification, labs are sent soils that are similarly processed and finely ground.

    **Proposed text:** Soil testing labs can elect to participate in quality assurance and quality control (QA/QC) certification programs that promote their data as high quality. For example, the North American Proficiency Testing (NAPT) Program of the Soil Science Society of America (SSSA) is one example of a program offered in the United States (U.S.), with over 130 NAPT certified labs. Participating labs are sent soil samples either quarterly or biannually and the data generated by each lab is subjected to a blind and double-blind statistical evaluation. Values within +/- 2.5 times the median absolute deviation (MAD) units of the median (S890 North American Proficiency Testing program oversight committee, 2020) are considered acceptable. However, labs receive soil already processed using the same methods.

Line 60: Root and rock fragments are not considered as part of the fine soil that is important for the biogeochemical processes. However, rocks and roots are still components of soils.

    **RESPONSE:** We will add "fine" to the revised manuscript.

Line 63-65: Do the authors have any reference that commercial labs do not remove coarse fragments. To my experience, research labs apply sieving and in general same sample preparation for agricultural and non-agricultural soils. Also, soil inventories prepare the fine soil prior to C measurements.

    **RESPONSE:** Unfortunately, we could not find a published study demonstrating that commercial labs do not remove coarse fragments. However, we believe we have support for this claim in the current manuscript. We presented results from a preliminary survey that showed over 70 % of the service labs surveyed use a mechanical flail grinder for the initial sieving step (L74; Supplemental Table S1). We also showed in Supplemental Table S2 that 5 of 8 labs used in our blind comparison sieve with a mechanical grinder and only 1 of 8 fine grind the sample beyond the 2 mm sieve (or in one case the 1 mm sieve). Because the whole bulk sample is poured into the grinder prior to falling over the 2 mm screen, it's safe to assume that coarse material is ground before being

removed. In the case that coarse fragments are picked out of the sample after a pass through the mechanical grinder, there still may be some that goes through the 2 mm screen initially.

Line 65-67: It is not clear to me why regenerative agriculture results in more coarse fragments in deeper soil. Also, the authors refer here rather to conservational land management rather that regenerative land management, which is a very broad and not well-defined term.

> **RESPONSE:** We agree that regenerative agriculture is a broad term and can involve many different types of management. We will link regenerative agriculture and deep-rooted perennials crops better as detailed below.

> **Original text (L65):** Compared to conventionally managed agricultural fields, coarse materials are more abundant in deeper soils in regenerative agricultural lands that include cover or perennial crops and grasses, thus it's important to consider how coarse materials in these soils may affect C estimation.

> **Proposed text:** Compared to conventionally managed agricultural fields, agricultural lands managed using a regenerative practice, like the addition of certain perennial crops (i.e., alfalfa), typically have more coarse materials deeper in the soil profile as more root biomass is incorporated at depth (Fan et al., 2016). Thus, it's important to consider how coarse materials in these soils may affect C estimation.

Line 77: Also here, the authors should be specific since it is considered as "fine soil"

> **RESPONSE:** We will add "fine".

Line 97: The authors should specify if near-infrared of mid-infrared regions.

> **RESPONSE**: Thank you for pointing this out. We will add "mid-".

Also, such approaches require a well-trained model based on large enough soil spectral library. This is a critical step for the quantification of soil C using chemometric approaches. Therefore, it follows a different concept compared to the other more direct methods.

> **RESPONSE:** Please refer to our response above.

Line 121: I would rather expect that the dual homogenisation by sieving to 8 followed by 2 mm would result in lower variability.

> **RESPONSE:** Yes, that is a good point, but we will keep our original hypothesis.

Table 1: is pH, %SOC and %SIC are measured with analytical replicates? The authors should add errors to the values.

> **RESPONSE:** Yes, % SIC and % SOC have replicates (n=5). We will add the standard deviation to % SOC, and % SIC and add a column for % TC with standard deviation reported.

Table 1 caption: How was pH measured and what are the texture classes applied? It is not clear what "Colorado State University following procedure P0" is. Please provide details of refer to Table 2 here.

**RESPONSE**: We agree so will provide more details for the pH and texture methods used and refer to Table 2 in the caption for the reference methods.

**Proposed text (L141):** The pH was determined using a 1:1 ratio of soil to deionized water. Texture was determined after shaking 40 g of soil in 5 % sodium hexametaphosphate solution for 18 hours, wet sieving sand > 53 μm, and using a hydrometer to determine silt and clay content. Texture classes were defined according to the soil texture calculator created by the Natural Resources Conservation Service U.S. Department of Agriculture.

Line 218-219: This is not very precise. It is not clear which model and was used and on which data it is trained.

Line 223-224: This is most likely attributed to the fact that the used model for the prediction based on the KSSL is developed with samples of similar degree of grinding. In the cited paper, Sanderman et al (2023) conclude that the model trained on fine milled samples was not well transferable on the coarser samples.

Therefore, the authors used a model that was trained for a certain milling. This makes this testing of grinding here not very useful. in comparison, Sanderman et al (2023) developed separate models for roughly 2400 samples of the KSSL. They conclude that a model that was trained with coarse samples and predicted coarse samples was performing similar to a model that was trained with fine soils and predicted fine soils. However, the transfer of models was not satisfying.

**RESPONSE:** We agree and very much appreciate this insight. Please refer to our response to the related general comment made above.

Line 231: Before it was mentioned that acid fumigation was only performed for P9. Here it reads like every sample was fumigated. Please clarify.

**RESPONSE:** We will clarify that acid fumigation was only used for the P9 (future Q2) procedure.

Line 241-243: How were CO2 and H2O interferences corrected?

**RESPONSE:** Thank you for this reminder. We will include a sentence describing how the $CO_2$ and $H_2O$ interferences were corrected as detailed below.

**Proposed text (L243):** A background of gold was scanned before every sample to correct for potential fluctuations and interference of $CO_2$ and $H_2O$.

Line 245-248: The authors should add more details regarding the predictions. Seybold et al (2019) developed PLSR based on the NSSC-KSSL. is this also used here? What do the authors mean with "respective geographical region"? Were the models local? Was there any spectral pre-processing like re-sampling, filtering, normalization or bassline correction?

**RESPONSE:** These are valid questions that we will address as detailed below. We will also include a supplemental table with each model summary.

**Current text (L245):** For predicting % TC, % SIC, and % SOC, spectra were trimmed from 4000 to 600 cm$^{-1}$. Models were built separately in the OPUS software (OPUS version 8.5, Bruker Optik GmbH 2020) for % TC, % SIC, and % SOC for each soil's respective geographical region using

the USDA NRCS National Soil Survey Center-Kellogg Soil Survey Laboratory (NSSC-KSSL) spectral library (Seybold et al., 2019).

**Proposed text:** For predicting % TC, % SIC, and % SOC, calibration models were built using the USDA NRCS National Soil Survey Center-Kellogg Soil Survey (NSSC-KSSL) spectral library coupled with partial least squares regression in the OPUS software (OPUS version 8.5, Bruker Optik GmbH 2020) as described in detail by Seybold et al. (2019). The calibration models were developed separately by soil property and geographical region (i.e., the state of Colorado (CO) was used as the boundary for making predictions with soils collected in CO). Spectra were trimmed to the mid-infrared region from 4000 to 600 cm$^{-1}$ and calibration spectra were mean centered with redundancies removed using principal component analysis and outliers removed based on ANOVA of residuals in OPUS. Details for the geographical boundaries, spectral pre-processing, R², and root mean square error of prediction for each model can be found in Supplemental Table S3.

**Table S3:** Summary for each model built using the United States Department of Agriculture Natural Resources Conservation Service National Soil Survey Center-Kellogg Soil Survey coupled with partial least squares regression in OPUS (OPUS version 8.5, Bruker Optik GmbH 2020) describing the soil property of interest for prediction, spectral library boundaries, spectral pre-processing for model optimization, and the validation model R² and root mean square error of prediction (RMSEP).

| Soil Property | Area Name | Spectral pre-processing | R² | RMSEP |
|---|---|---|---|---|
| Total Carbon | Colorado | First derivative + Vector normalization (SNV) | 0.9506 | 0.433 |
| Total Carbon | Wyoming | First derivative + Vector normalization (SNV) | 0.9543 | 0.442 |
| Total Carbon | Iowa or Nebraska | First derivative + Vector normalization (SNV) | 0.9625 | 0.309 |
| Total Carbon | Kansas | First derivative + Vector normalization (SNV) | 0.984 | 0.44 |
| Inorganic Carbon | Colorado | First derivative + Straight line subtraction | 0.9899 | 0.834 |
| Inorganic Carbon | Wyoming | First derivative | 0.9925 | 0.761 |
| Inorganic Carbon | Iowa or Nebraska | First derivative + MSC | 0.9768 | 1.56 |
| Inorganic Carbon | Kansas | First derivative + Straight line subtraction | 0.9959 | 0.798 |
| Organic Carbon | Colorado | First derivative + Vector normalization (SNV) | 0.963 | 0.398 |
| Organic Carbon | Wyoming | First derivative + Vector normalization (SNV) | 0.9604 | 0.318 |
| Organic Carbon | Iowa or Nebraska | First derivative + Vector normalization (SNV) | 0.9553 | 0.318 |
| Organic Carbon | Kansas | First derivative + MSC | 0.9564 | 0.257 |

Line 270: "External service labs provided values for % TC, % SIC, and % SOC." can be removed.

**RESPONSE:** We will remove this sentence.

Line 271: Looking at Table S3, it seems not fair to just select the extremes here. Most differences are rather lower. It is hard to tell from the table. Maybe boxplots per soil with different symbols for the labs would be easier to read. Anyway, the authors should also mention the range of differences and not only the extremes.

**RESPONSE:** This is a good suggestion, thank you. We only reported on the extremes in the current version because that's a big take away with this dataset, but it's true that, in some cases, the distribution is much tighter. We will add to the text to include more results for the blind comparison as detailed below and update Fig. 1 to present the data more clearly. In the revised manuscript Fig. 1 will be Fig. 2 as proposed below.

[Figure]

**Figure 2:** The distribution of total carbon (TC; panel a) soil inorganic carbon (SIC; panel b) and soil organic carbon (SOC; panel c) concentrations from eight service soil testing laboratories and Colorado State University (CSU). Box plots report the median, first and third quartiles for values from all soils (field moist and air-dried) analyzed at service soil testing laboratories (brown boxplot) and CSU (grey boxplot; n=5). Whiskers extend to the upper and lower data point that are within 1.5 times the interquartile range. For soils B, C, D, and J, two samples were sent to each external

lab, one air-dried and one field moist (n=16). One sample from soil H was sent to each lab (n=8). Refer to Table 1 for a description of the soils, Figure 1 for Reference (CSU) methods, and Supplemental Table S2 for external service soil testing laboratory methods.

**Proposed text (L274):** However, in some cases, labs reported the same or similar values either air-dried or field moist. For example, Lab VII, reports no difference in % SOC while Lab I only detected a 0.01 % difference between the air-dried and field moist samples sent from soil B. Lab VI reported differences of < 0.1% TC for all soils sent while Lab I reported differences in SOC of < 0.1 % for all soils.

Line 284: Yes, it is an astonishing range of measured values between labs. It is also surprising that the reference measurement (CSU lab) shows a large variability of soil B and H. These are two soils with high pH. I wonder if this could be an effect of the carbonate removal. What is your explanation for large differences between the five analytical replicates? Additional, the external labs did not measure in replicates?

**RESPONSE:** Yes, we realize that there is notable variability within the CSU lab for these two soils. These two soils have the highest variability by far, so we wanted to send them to the external labs for comparison. However, we will note that when the distribution of CSU's 5 reps is compared to the distribution across the external labs, the external labs (viewing each data point like a replicate) have a larger distribution. Given that soils were homogenized for subsamples using the same method for CSU and for the external labs, we attribute this to differences in soil processing used across labs. We will add our explanation for the variability and compare it to the variability across the external labs as detailed below:

**Revised text (L403):** However, we observed notable variability across the CSU reference for soils B and H. We speculate that the higher variability in % SOC in soil H is due to the measured values for % SIC using the pressure transducer since % TC variability is very low and % SOC is calculated by % TC - % SIC for the CSU (R) method. The presence of substantial amounts of fine root and some SIC in soil B (irrigated pasture) is most likely the reason for high variability using the CSU R method. By sending external labs the two soils with the highest variability, we were able to confirm the expectation that high % SIC and high fine root material contribute to higher variability in SOC data. Additionally, we can attribute higher variability to differences in processing methods as, despite a notable distribution in soils B and H from CSU, the distribution across all external labs is much larger (Fig. 2).

We regret that we did not have enough soil to send 5 replicates of field moist and air-dried soils to each external lab. That was an unfortunate oversight. We describe this in the current manuscript L150.

Line 288: Why are the no coarse materials at all in soil L for the P3?

**RESPONSE:** Good observation. There was no coarse material collected from any of the replicates using the S3 procedure, meaning that it was all ground into the sample and considered fine soil.

Line 305-307: This relationship seems to be mainly driven by the on P3 point at 0.8 difference in plant material and 1 STD %SOC (right corner). This is in general a very weak correlation and might not add much when the one point is considered as an outlier.

**RESPONSE:** We discussed this among the co-authors as a potential issue prior to the preprint so will take your point of view into consideration as well and remove Fig. 3 in addition to any text referring to the figure. Thank you for your input.

Line 323-324: Do the author mean a relationship to SOC, similar to the plant material in Fig. 3?

**RESPONSE:** Yes, we will clarify that in the revised manuscript.

Figure 4 and results section: This paper is mainly about errors that are important for the SOC because this will be of interest for the C markets. Therefore, I wonder if Figure S3 with the SOC differences between soils and methods should be the main figure in the manuscript and the current Figure 4 could more to the SI. This might need a restructuring of the section as well.

**RESPONSE:** Yes, that's true. We will replace Figure 4 with the proposed figure below in the main text and revise the text accordingly.

**NEW FIGURE:**

[Figure]

**Figure 4:** The difference (Δ) in % soil organic carbon (SOC) compared to the reference (R) mean value for all sieving procedures including R as described in Figure 1. Box plots report the median, first and third quartiles. Whiskers extend to the upper and lower data point that are within 1.5 times the interquartile range. Letters indicate the different soils, as described in Table 1, which are arranged on the x-axis by proportion of rock material removed with the R sieving procedure.

Line 339: Significances are shown in Table S6?

> **RESPONSE:** Yes, the referee is correct. We will add that to the text for readers.

Figure 6: X axis label, colour and legend are redundant.

> **RESPONSE:** We will improve this figure in the revised manuscript.

Line 396-397: Here the focus is on SOC for the C market.

> **RESPONSE:** The referee is correct. We will clarify SOC instead of C, in the revised manuscript.

Line 398-399: Please see my comment regarding the variability on CSU lab for some soils. This is also concerning. Here it would be good to have replicates from the individual labs.

> **RESPONSE:** Please refer to our response above.

Line 460: This would be a very interesting aspect of the manuscript. The C market needs stocks of C and not concentrations alone. Therefore, the effect of removed or not removed coarse fragments would be most significant. Even the calculation of SOC stocks includes large uncertainties and this would add up with the method uses (e.g. Poeplau et al. 2017). The authors do back on the envelope calculations later in the implications section. Would it be possible to discuss the stock effects even more by estimating the stock differences here for all methods using the soils bulk densities?

> **RESPONSE:** We appreciate the enthusiasm about this aspect we added to the manuscript. While we would love to honor this suggestion, we were unable to calculate bulk density because we did not get an accurate volume of soil when we collected it by shovel. For that reason, we chose to assume a bulk density of 1 g/ cm$^3$.

Line 465-468: Yes, plant material would be low in mass but might be important in volume and thus could have an impact of stocks as well.

> **RESPONSE:** That's a good point but we do not consider the volume of roots in our bulk density calculations as described in Poeplau et al. (2017). Only the mass is considered.

Line 567: This should be Fig. S9

> **RESPONSE:** Thanks, we will correct the figure number.

---

## Author Response (AR3)

Dear Authors,

both reviewers have agreed that the manuscript improved considerably and can be published if a few corrections are implemented:

First, figure 7 needs to be overhauled, a 1:1 line added, and the x and y axes need to have the same range. Both reviewers state that the way that it is presented, deviations from the 1:1 line are not clearly visible.

Secondly, reviewer 1 recommends to state the limitations of the MIR/FTIR method even more clearly. These are mainly the missing libraries in some regions of the globe and that it is questionable if MIR has the precision to detect changes in SOC at the field scale (much harder than predicting SOC across a range of soils). Consequently, reviewer 1 recommends to temper the claims about the general applicability in the abstract, main text and conclusion.

**RESPONSE:** Thank you for sharing the referee comments and for giving us the opportunity to make the suggested minor revisions for resubmission. Figure 7 has been updated to address all the referees' comments and each specific comment has been addressed as detailed below.

[Figure]

**Figure 7:** All quantification methods for % soil total carbon (TC), % soil inorganic carbon (SIC), and % soil organic carbon (SOC) plotted against the reference method where Q1 is predictions using Fourier transformed infrared spectroscopy, Q2 is acid fumigation, and Q3 is loss on ignition as described in detail in Fig. 1. The dashed line represents a 1:1 relationship.

To expand on the limitations of FTIR spectroscopy, we have revised L34-35 in the abstract to read "Moreover, we show promising results using FTIR spectroscopy coupled with predictive modeling for estimating % TC, % SIC, and % SOC in regions where spectral libraries exist." We have also added text as detailed below.

**Proposed text (L577):** Additionally, the KSSL library used in our study was representative of the geographical region for our sample set. The effectiveness of FTIR coupled with predictive modeling depends on the accessibility of spectral distribution within the geographical area of interest. Projects may be limited by the spectral libraries available.

**Proposed text (L596):** Moreover, as the use of FTIR gains traction, laboratories need to be aware that model transfer from a large spectral library (like the KSSL) may be problematic if the instrumentation used to analyze the soils does not match the instrument used to build the spectral library (Safanelli et al., 2023).

**Proposed text (L641):** We also encourage future research to better understand if the precision of FTIR predicted SOC measurements is sufficient to detect changes in SOC at the field scale.

Specific comments:
Line 67-68: It might be worth to mention here that international recommendations exist. For example, the ISO 23400:2021 that aims to standardize the sampling, measurement (e.g. dry combustion) and calculations. This is also partly discussed in Meurer et al. (2024) for European countries.

ISO. (2021). ISO 23400:2021(E), Soil quality-Guidelines for the determination of organic carbon and nitrogen stocks and their variations in mineral soils at field scale.

Meurer et al. (2024): 8

>    **RESPONSE:** Thank you for this suggestion. We have added the following text as detailed below.

>    **Proposed text (L67):** And while several standards exist that guide methodologies for soil analysis of SOC (e.g., the International Organization for Standardization (ISO) 23400:2021(en)), divergence of methods occurs as the guidelines are not followed by every lab, especially for soil preparation procedures prior to elemental analysis.

Line 110: It is rather that the MIR spectroscopy is coupled with predictive modelling, which needs to be well trained. Please rephrase.

>    **RESPONSE:** Thank you. We have reworded the sentence to read "…Fourier-transformed infrared spectroscopy (FTIR; Reeves, 2010; Goydaragh et al., 2021) where samples are scanned in the mid-infrared region. The produced spectra are then coupled with predictive models that must be well trained to produce accurate soil C estimates."

Line 165: To be complete, it would be good to have this information site specific. Can it be added for each soil in Table 1? At least the land use during sampling.

> **RESPONSE:** We have added a column with the land use and added text to the caption accordingly.

Line 173: If I am not missing it, CSU is introduced as an abbreviation in Line 187.

> **RESPONSE:** Good catch. We will introduce the abbreviation at the first use of it.

Line 256: The NRCS-KSSL is the basis for the model used here in OPUS.

> **RESPONSE:** Correct. We have added "and models" to L259.

Figure 7: To make it easier to read, a 1:1 line would be helpful here.

> **RESPONSE:** We agree and have added a 1:1 line as illustrated above.

Line 453-453: Yes, selecting roots after sieving can be important here. I wonder how you would standardize this. To my experience it is very hard. One can spend hours for one sample. What is your opinion here to get hands on this in a mor systematic way?

> **RESPONSE:** Yes, the authors agree that it is difficult. The best approach may be to standardize a time for picking roots that is site (soil) dependent. However, the defined time needs to be within reason. This is why we usually only 2 mm sieve a representative subsample of the whole soil based on analytical needs so we can scale the proportion of fine earth after 2 mm sieving up to the whole sample (for bulk density) without spending hours picking roots. We will add these suggestions to the revised manuscript.

> **Proposed text (L455):** In soils with a high density of fine root material, extra time may be necessary to adequately remove roots using tweezers. The time could be standardized by site (soil) so that all samples receive the same treatment. It is also worth noting that a representative subsample of the whole soil could be 2 mm sieved to eliminate the time needed to remove coarse materials.

Line 498-499: The automatic machines were less efficient with clayey soils? This is not clear and how you can make this statement here, did you test? In general, I am not sure what this adds to the discussion.

> **RESPONSE:** Yes, we did perform an internal test at the Soil Innovation Lab and concluded that the automated sieving machine did not save time sieving heavy clay soils because the aggregates did not break well. We included this in the discussion based on another referee's suggestion and will retain the text because it is important that readers know automated sieving machines are on the market and could be a potential option but have not been tested in a formal study (to our

knowledge). However, we have added "...after testing one machine internally at CSU" to be clearer.

Line 542-548: That is very interesting and good that you are able to emphasize the issue to SOC volatilization.

**RESPONSE:** Thank you. We agree. It is difficult to find publications related to this issue.